# Bridging Brains and Concepts: Interpretable Visual Decoding from fMRI with Semantic Bottlenecks

**Sara Cammarota**[*]
Department of Biomedicine and Prevention
University of Rome, Tor Vergata
Viale Montpellier, 1 – Rome (IT)
`sara.cammarota@uniroma2.eu`

**Matteo Ferrante**[*]
Tether Evo
Department of Biomedicine and Prevention
University of Rome, Tor Vergata
`matteo.ferrante@tether.io`

**Nicola Toschi**
Department of Biomedicine and Prevention
University of Rome, Tor Vergata
Martinos Center For Biomedical Imaging
MGH and Harvard Medical School (USA)
`toschi@med.uniroma2.it`

## Abstract

Decoding of visual stimuli from noninvasive neuroimaging techniques such as functional magnetic resonance (fMRI) has advanced rapidly in the last years; yet, most high-performing brain decoding models rely on complicated, non-interpretable latent spaces. In this study we present an interpretable brain decoding framework that inserts a semantic bottleneck into BrainDiffuser, a well established, simple and linear decoding pipeline. We firstly produce a $214-$dimensional binary interpretable space $\mathcal{L}$ for images, in which each dimension answers to a specific question about the image (e.g., "Is there a person?", "Is it outdoors?"). A first ridge regression maps voxel activity to this semantic space. Because this mapping is linear, its weight matrix can be visualized as maps of voxel importance for each dimension of $\mathcal{L}$, revealing which cortical regions influence mostly each semantic dimension. A second regression then transforms these concept vectors into CLIP embeddings required to produce the final decoded image, conditioning the Brain-Diffuser model. We found that voxel-wise weight maps for individual questions are highly consistent with canonical category-selective regions in the visual cortex (face, bodies, places, words), simultaneously revealing that activation distributions, not merely location, bear semantic meaning in the brain. Visual brain decoding performance are only slightly lower compared to the original BrainDiffuser metrics (e.g., the CLIP similarity is decreased by $\leq 4\%$ for the four subjects), yet offering substantial gains in interpretability and neuroscientific insights. These results show that our interpretable brain decoding pipeline enables voxel-level analysis of semantic representations in the human brain without sacrificing decoding accuracy.

## 1 Introduction

Decoding the contents of the human mind from neural activity is one of the main challenges of contemporary neuroscience. Brain decoding refers to the attempt to classify, retrieve or reconstruct the stimuli that elicited a certain neural response, in an effort to translate the language of the human

---

[*]Equal contribution.

39th Conference on Neural Information Processing Systems (NeurIPS 2025).

mind into a representation of the external world (visual scenes, linguistic input, music or other experiential domains) [48, 25, 24, 23, 12, 26, 35, 39, 17, 19, 4, 8, 9, 14, 46].

Recent progress in the field of Artificial Intelligence, together with increased availability of large, high-quality open neuroimaging data [2, 7, 22, 28, 36, 29], have significantly propelled the field forward[39]. Very quickly it became possible to decode visual [41, 10, 44, 18, 20, 31, 21], language related [3, 13, 46], semantic[15] and music[14, 5, 16] content from brain activity, and state of the art models are capable of doing so even when the neural correlates are measured with non-invasive techniques such as functional magnetic resonance imaging (fMRI), which infers neural activity by measuring the blood oxygen level dependent (BOLD) signals across the brain. Even though fMRI is limited by low temporal resolution, a consequence of the slowly evolving hemodynamic response function underlying BOLD signal, it presents substantial advantages: its spatial resolution is extremely precise, it allows for whole brain coverage and it is non invasive. These factors led fMRI to become an indispensable tool for cognitive neuroscience, with fMRI datasets being made accessible and well performing encoding and decoding models being proposed in recent literature. One of the most widely used datasets for brain decoding is the Natural Scenes Dataset (NSD) [2], the product of a large-scale data collection initiative following the principles of "intensive fMRI" [27]. This approach shifts the focus from traditional "wide fMRI" studies (many subjects, few trials) to "deep" or "intensive fMRI" (few subjects, many trials), enabling the development of subject-specific models that leverage machine learning and facilitate experiments supporting a broad range of neuroscientific hypotheses. For these reasons, we selected the NSD as the foundation for our work.

On visual brain decoding specifically, a majority of works have converged on a dual-stream architecture combined with a generative model: a semantic stream predicts high-level representations of the stimuli while a structural stream maintains structural features of the image, and a generative model uses these predictions to produce the final stimuli reconstruction [32, 37, 41]. These models have proven very effective and have demonstrated very high performance, especially on the NSD dataset. Deep learning approaches like these have revolutionized the field; however, a striking result is that simple linear regressions from brain activity to latent semantic representations (such as those obtained by CLIP) can achieve good, highly-generalizable performance [38, 42]. This result may be explained by the simultaneous action of macroscopic neural dynamics, such as time and space averaging and noise, which could mask the nonlinearities of smaller scales. At the macroscopic fMRI scale, the representation of concepts in the human brain may resemble the vector space of self-supervised large models like CLIP, effectively making linear models very efficient, other than robust, easy to train and potentially interpretable. Nonetheless, a linear mapping between brain voxels and the entangled CLIP feature space still doesn't allow for straightforward interpretation.

Here we introduce an interpretable brain-decoding framework that bridges this gap. Inspired by a recent related study on language and the idea of factorizing linear models in concept-specific components, we propose an approach to map brain activity into a human-readable embedding space that serves as middle ground between neural correlates and the semantic embeddings of CLIP. Each dimension of this intermediate space corresponds to a visual concept (such as *people*, *motion*, *words*), allowing us to gain interpretability on the input space, eventually advancing our knowledge of the brain. We insert this framework within BrainDiffuser[41], a well-established reconstruction pipeline, allowing us to benchmark our results and ultimately to test whether interpretability impacts image reconstruction performance.

**Related Work**   Our work falls within the umbrella of the recent field of brain decoding literature, which demonstrated that it is possible to decode images, videos, language and music from brain activity measured via fMRI [41, 10, 44, 3, 13, 46, 15, 14, 5, 16, 9]. The central idea of this work is that the representations of external stimuli in the human brain and their latent counterparts in large-scale models share some similarities, and with sufficient {stimulus, neural activity} pairs the two representations could be related to each other. Two recent pieces of literature, in particular, align with our study[6, 34]. First, [6] elegantly showed that language brain encoding models can achieve good performance when non-interpretable sentence embeddings are replaced by interpretable vectors whose dimensions represent the answer to different questions (e.g. "Does the sentence describe a relationship between people?", "Is the sentence grammatically complex?"). This way of crafting the interpretable latent space really resonates with the way we design our interpretable embeddings for images. Second, the BrainBits study [34] demonstrated that it is possible to impose bottlenecks in a visual stimuli decoding pipeline without losing much performance, suggesting that

brain representation could be more compact and lower dimensional compared to the high-dimensional embeddings produced by large-scale multimodal models such as CLIP. While a vast majority of literature focuses on maximizing decoding performance leveraging several non-interpretable spaces, here we take inspiration from[6, 34] and adopt a slightly different perspective: given the high performance reached by visual brain decoding models, we leverage the power of these predictive models to probe the structure of conceptual representations in the brain space.

**Contributions** The contributions of this work are: (i) we introduce an interpretable semantic bottleneck, interpretable *by design*, that preserves the information required for high-quality image reconstruction while providing each dimension with a clear meaning; (ii) we propose a pipeline to produce robust interpretable embeddings from a set of images, starting with the generation of a number of questions about the NSD carried out by an LLM, GPT-4o, followed by the answering of these questions by another model, BLIP-2; (iii) we focus on the semantic stream rather than the structural stream, decomposing the *brain to semantic* mapping into two separate maps: brain to interpretable space, and interpretable space to semantic CLIP space. This yields voxel-level insights that are easily visualized and compared to known activation patterns in the human brain. (iv) we show that the proposed model produces stable, anatomically plausible *concept maps* across subjects; we furthermore observe that the distribution of voxel-pattern strength (not only the location of the activations) carries important information.

**Main findings** The derived interpretability maps align well with canonical regions in the visual cortex significantly and for several concepts. The results also highlight the presence of distributed networks of voxel activations, reinforcing the idea that high-level representations are encoded in our brains by large patterns of coactivations rather than small, isolated hotspots. Our approach combines good reconstructing accuracy with built-in transparency, offering a new way to fine-grained studies of semantic representations in the human visual cortex.

## 2 Methods

**Data** We used the Natural Scenes Dataset (NSD)[2], a deep fMRI dataset encompassing eight healthy adult subjects who performed a continuous recognition task on thousands of images from the COCO dataset. The data were acquired in high-resolution with ultra-high-field (7T) strength. NSD data can be requested at `https://naturalscenesdataset.org/`. To facilitate comparison with other works employing the same dataset, we considered in our analysis only those subjects who completed all trials (subj01, subj02, subj05, subj07). We obtained a training set of 8859 images and 24980 fMRI trials per subject, while the test set included 982 images and 2770 fMRI trials per subject. Because stimuli were presented to subjects up to three times, the corresponding fMRI trials were averaged. The fMRI signal, in 1.8 mm resolution, was masked using the NSDGeneral region-of-interest mask (this includes many areas of the visual cortex), which reduced spatial dimensionality to approximately 15000 voxels per subject. On the temporal dimensionality, we used NSD-supplied beta weights from a general linear model with fitted hemodynamic response functions; this condensed each voxel's response to a single value per stimulus. Full acquisition and preprocessing details are available in the original NSD publication [2].

**Classical visual decoding pipeline** To develop our interpretable pipeline, we build on a classical brain decoding method for images, BrainDiffuser[41]. We choose BrainDiffuser because of its strong performance, modular architecture, and simplicity of implementation. Its clarity and flexibility make it well-suited for integrating our interpretable semantic bottleneck without introducing additional confounds. BrainDiffuser is a well-established brain decoding framework that includes a two-stage scene reconstruction pipeline encompassing different latent representations of the images (VDVAE[11] and CLIP[43]). The VDVAE is a hierarchical variational auto-encoder that stacks a series of conditionally dependent latent layers, allowing each layer to capture a different aspect of image structure, from fine details to coarse global information. The original BrainDiffuser exploits the first 31 of the 75 VDVAE layers. At training time, images are fed to the pretrained VDVAE encoder and latent variables corresponding to these first 31 layers are concatenated. A ridge regression model is trained to predict these concatenated VDVAE latents $\mathbf{z}$ from fMRI activations. At test time, the trained regressor is used to predict $\tilde{\mathbf{z}}$ from the new fMRI. $\tilde{\mathbf{z}}$ latents are then fed to the pretrained VDVAE decoder, resulting in a $64 \times 64$ low-level reconstruction of the image, which is structurally

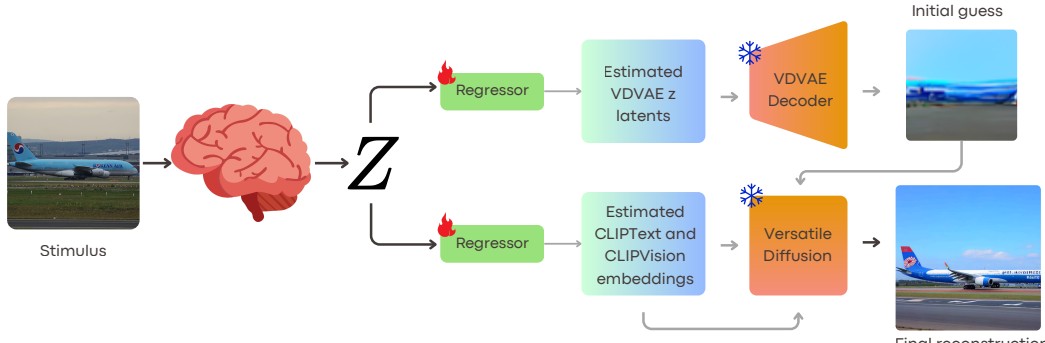

Figure 1: The BrainDiffuser model reconstructs stimuli from fMRI data. A first, structural stream (top) is trained to estimate the representations of a VDVAE while a second, semantic stream (bottom) estimates CLIP-Text and CLIP-Vision latent features $\widetilde{h}$ from neural data $Z$. Both estimations are done with linear models. The VDVAE latents are fed to the VDVAE decoder to produce a first initial guess of the image, which is given as input to Versatile Diffusion together with the estimated CLIP-Text and CLIP-Vision vectors to obtain the final reconstruction.

similar to the original picture, but lacks fine semantic detail and is still unrecognizable. This serves as an initial guess for the second stage of the BrainDiffuser. The second reconstruction stage leverages Versatile Diffusion (VD) [49], a latent diffusion model which allows conditioning the generation process on both text and image features to guide the reverse diffusion process. The reverse diffusion of VD can also be initialized with latent variables obtained by an image. Stage 2 of the BrainDiffuser consist in the training of two additional regression models: the first one maps fMRI patterns to CLIP-Vision embeddings of the stimuli, and the second one maps fMRI data to CLIP-Text features (obtained from the captions of the COCO dataset corresponding to each NSD stimulus).

At testing time, also the initial guess is used: it is encoded by the AutoKL Encoder of the pretrained Versatile Diffusion model and noise is added to the obtained latent vector in 37 steps; the so obtained noisy vector is then fed as initialization to the diffusion model and denoised for 37 steps, while conditioning on the predicted CLIP-Text and CLIP-Vision features $\widetilde{\mathbf{h}}$. The result of the diffusion process is then fed to the AutoKL Decoder of the pretrained Versatile Diffusion model and the final image reconstruction is obtained. A schematic representation of the BrainDiffuser is available in Figure 1. For further details, we refer to the original publication[41].

Notably, in the second stage of the original BrainDiffuser brain activity $Z$ is mapped into the estimated CLIP embeddings by means of a linear model $\widetilde{\mathbf{h}} = ZW$ where $W$ has shape ($n_{\text{voxels}}$, $\text{CLIP}_{\text{dimension}}$). Because the CLIP embeddings are not interpretable, it is difficult to understand the meaning of the model weight matrix $W$.

**Interpretable space** To understand how brain activity is converted into the representations given by the CLIP embeddings, we factorize the decoding weight matrix $W$ in two separate matrices $A$ of size $n_{\text{voxels}} \times a$ and $B$ of size $a \times \text{CLIP}_{\text{dimension}}$, such that $W = AB$. The core idea is that we can design an $a-$ dimensional, semantically interpretable latent space $\mathcal{L}$ so that $A$ : brain activity $\rightarrow \mathcal{L}$ and $B : \mathcal{L} \rightarrow$ CLIP space. This decomposition allows for direct inspection of $A$, whose entries reveal the individual contribution of each brain voxel to every latent dimension. Therefore, the factorization exposes a linear map onto semantic dimensions, enabling neuroscientific interpretation.

To design a meaningful latent space $\mathcal{L}$ we adopt a two-step pipeline. First we leverage GPT4-o [40] to create a set of non-overlapping questions with binary answer (yes or no) that well describe the images of the NSD dataset (e.g. "Is there a person in the image?", "Is the image taken outdoors?", "Is there more than one subject?" etc.). This was done by providing the LLM with the captions for all images and the prompt "Given the following set of captions, each representing an image, generate a set of 256 binary questions that are suitable to well describe the content of the images. Generate questions that are diverse and non-overlapping, and that describe as completely as possible all the images. These should maximize your information about the image content. The prompt asks for 256 questions. This number was set with the idea in mind of imposing a bottleneck that reduces

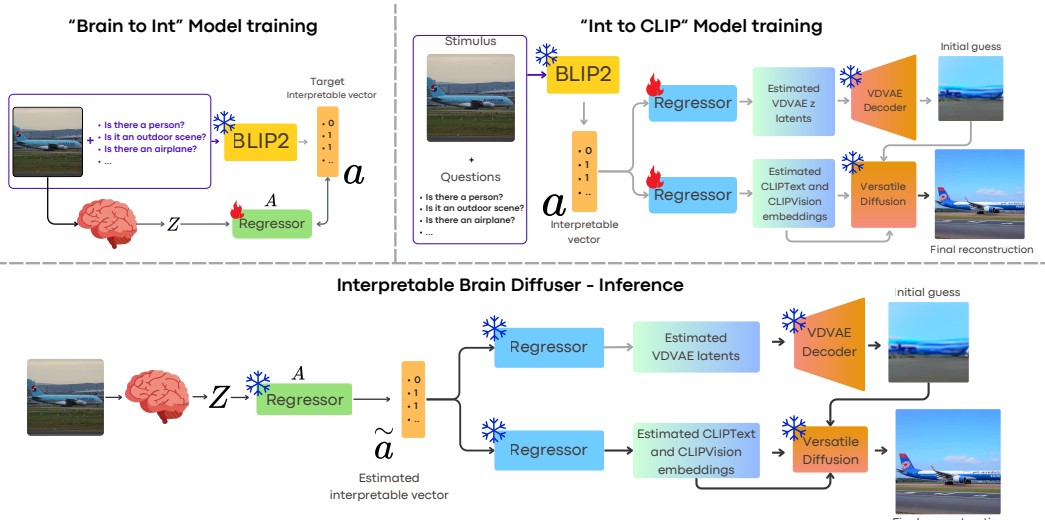

Figure 2: Overview of the different models. Top row: (1) "Brain to Int" Model training: training pipeline of the model mapping from brain activity → intepretable space $\mathcal{L}$. Each stimulus image is fed to BLIP-2 along with a fixed set of $214$ yes/no questions. The model's answers form the concept vector $\mathbf{a}$, our interpretable target space. The "Brain to Int" model learns to reconstruct these vectors from brain activity; (2) "Int to CLIP" Model training: training pipeline of the model mapping from $\mathcal{L} \to$ CLIP/VDVAE space. This model learns to map brain activity to VDVAE and CLIP embeddings which are used to generate the reconstructed image. Bottom row: "Interpretable BrainDiffuser - Inference": the whole inference pipeline of our interpretable model. Neural data $Z$ is first mapped to the interpretable space $\mathcal{L}$ via the "Brain to Int" model and then mapped to CLIP/VDVAE space via the "Int to CLIP" model). The final image is generated via Versatile Diffusion, using the estimated CLIP/VDVAE embeddings.

the dimensionality between 20% and 30% of the original CLIP dimensionality (768), knowing from a previous literature result that a bottleneck of 50 dimension is already able to retain most of the performance [34]. After manual inspection of LLM generated questions, removing duplicate questions, we select a reduced set of 214 questions. Each dimension of $\mathcal{L}$ corresponds to the answer of a specific question. The complete set of questions is available in the Supplementary Materials. Later, BLIP-2[30] was employed to answer the $214$ questions sequentially for each image in the dataset with the following question-specific prompt: "Question: [*one question, for example 'is the subject smiling?'*] Answer:". For every image, the model was queried 214 times, once per question, producing a binary ("Yes"/"No") answer for each. The answers were converted to binary values (1 for "Yes", 0 for "No"), resulting in a 214-dimensional interpretable vector representation for each image.

Figure 2 is a scheme of the interpretable brain decoding pipeline.

We verify that $\mathcal{L}$ preserves CLIP-level semantics by learning a ridge-regression alignment to CLIP-Vision and measuring top-$k$ image-retrieval accuracy. The full protocol and results appear in Appendix 6.1.

**Reconstructing images from interpretable embeddings**     We first estimate the linear map $B : \mathcal{L} \to$ CLIP by duplicating the BrainDiffuser architecture and training a Ridge regression[1] to reconstruct the images from ground-truth interpretable embeddings (instead of brain activity). We used 5-fold cross-validation to determine the best values of the regularization coefficient $\alpha$ over logarithmically spaced values in the interval $10^{-3} \leq \alpha \leq 10^{4}$. This latent-to-image model has two purposes. First, it verifies that the designed latent space is sufficiently informative, as high reconstruction quality indicates that the latent space encodes important properties of the NSD images. Moreover, this model is subsequently frozen and applied to the interpretable embeddings inferred from brain activity to generate the final reconstructed images.

---

[1]We use Python library Himalaya Ridge regression with the standard svd solver.

**Learning the map from brain activity to $\mathcal{L}$** We fit a ridge regression model[2] to estimate the normalized coordinates (because $B : \mathcal{L} \rightarrow$ CLIP expects normalized inputs) of the interpretable space $\mathcal{L}$ from brain activity. 5-fold cross-validation was used to determine the optimal values of the regularization coefficient $\alpha$ over logarithmically spaced values in the interval $10^{-6} \leq \alpha \leq 10^6$. The learned map $A$, one per subject, is interpretable by design: each $i - $ th column encodes the contribution of every brain voxel to the $i - $ th coordinate in $\mathcal{L}$, i.e. the answer to each $i - $ th binary question.

**Evaluation** We assess the quality of the images generated by our interpretable model and we compare them to the classical, non-interpretable, BrainDiffuser based on low- and high-level metrics. Low-level metrics include PixCorr, SSIM, MSE, Cosine Similarity, 2-way accuracy in AlexNet latent space, while high-level metrics include 2-way accuracy in InceptionV3 and CLIP latent spaces, correlation distance in EfficientNet and SwAV spaces, as well as 50-way-top-1 Accuracy using ViT-H/14 (1000 repetitions per image), replicating the original publication evaluation [41].

**Visualization** Visualization of brain regions involved most in the estimation of each dimension of $\mathcal{L}$ was achieved by first mapping each Region Of Interest (ROI) to the actual anatomical space of each participant. We then register these coordinates to the MNI-152 standard space, providing a common frame for group level analyses and comparison with canonical atlases. We compared the resulting maps to visual cortex regions knowingly related to concepts such as bodies, faces, words and places. The analysis was performed in Python using the Nilearn neuro-imaging library [1]. For final visualizations, we kept only the top $4\%$ of most influential voxels and discarded any cluster smaller than $100$ voxels to suppress noise. The decision to focus on the top $4\%$ of most influential voxels was made empirically, guided by two considerations. First, this threshold yields a voxel count that is comparable to that of well-characterized functional regions in the dataset, such as face-, body-, place-, and word-selective areas, each comprising a similar number of voxels[2]. Second, this proportion provides a clear and interpretable visualization.

All experiments and model training were carried out on a server equipped with 8 NVIDIA H100 GPUs, 2 TB of RAM, and 256 CPU threads. The extraction of interpretable latent embeddings with BLIP-2 took approximately 24 hours per subject and the training of the whole interpretable model took less than 30 minutes per subject. Inference time for all models is approximately 3 seconds per decoded image. Code is available at `https://github.com/SaraCammarota/Bridging-Brains-and-Concepts`.

## 3 Results

Tables 1 and 2 present the results of the interpretable model evaluation compared to the original BrainDiffuser and to the intermediate model "Int to CLIP". This last model maps the true interpretable vectors to the CLIP-Text and CLIP-Vision embeddings and serves as a check that the designed interpretable embeddings are good enough for image reconstruction. The tables report, respectively, low- and high-level metrics on the decoded images of the test set for the four subjects.

Results show that our interpretable model (Int-BD) retains good reconstructing performance (e.g., on average we report $-4\%$ on CLIP score and $-8\%$ on 50-way-top-1 Accuracy with respect to the original BrainDiff), confirming that interpretability does not affect excessively the reconstructing accuracy. It is important to notice that, even if *Int to CLIP* obtains the best high-level performance across all models, this is not a brain decoding model and only serves as an indication of how well $\mathcal{L}$ represents NSD images. Its superior accuracy stems from being the only model to bypass the noisy fMRI data, mapping directly the true interpretable embeddings to the CLIP space.

Figure 3 reports a collection of image reconstructions obtained with the three pipelines for different subjects. Images reconstructed by our interpretable model show visual fidelity and similarity to the ground truth images, comparable to the ones achieved by the original, non-interpretable BrainDiffuser.

Figure 4 displays the column entries of the interpretable matrix $A$, mapped to the MNI-512 space, for a subset of the interpretable embedding dimensions. We compare the activated voxels to the standard ROIs related to bodies, faces, places and text/words. The so-obtained activation maps exhibit a good similarity with standard concept-related regions.

---

[2]See Footnote 1.

Table 1: Quantitative analysis of reconstructed images for all subjects. Here are reported low-level image metrics. Upward arrow (↑): higher is better; downward arrow (↓): lower is better.

| Subject | Model | PixCorr ↑ | SSIM ↑ | MSE ↓ | Cosine Sim. ↑ | AlexNet(2) ↑ | AlexNet(5) ↑ |
|---|---|---|---|---|---|---|---|
| Subj01 | BrainDiffuser | 0.285 | 0.359 | 0.119 | 0.801 | 0.948 | 0.966 |
| | Int-BD | 0.153 | 0.320 | 0.140 | 0.767 | 0.795 | 0.869 |
| | Int to CLIP | 0.156 | 0.309 | 0.148 | 0.757 | 0.813 | 0.894 |
| Subj02 | BrainDiffuser | 0.241 | 0.355 | 0.126 | 0.789 | 0.931 | 0.956 |
| | Int-BD | 0.149 | 0.319 | 0.141 | 0.764 | 0.783 | 0.866 |
| | Int to CLIP | 0.155 | 0.311 | 0.149 | 0.755 | 0.819 | 0.897 |
| Subj05 | BrainDiffuser | 0.214 | 0.344 | 0.133 | 0.779 | 0.910 | 0.950 |
| | Int-BD | 0.154 | 0.320 | 0.140 | 0.765 | 0.798 | 0.878 |
| | Int to CLIP | 0.154 | 0.312 | 0.149 | 0.758 | 0.808 | 0.894 |
| Subj07 | BrainDiffuser | 0.213 | 0.345 | 0.134 | 0.781 | 0.901 | 0.940 |
| | Int-BD | 0.142 | 0.317 | 0.143 | 0.761 | 0.787 | 0.861 |
| | Int to CLIP | 0.160 | 0.311 | 0.148 | 0.756 | 0.817 | 0.895 |

Table 2: Quantitative analysis of reconstructed images for all subjects with high-level metrics. Upward arrow (↑): higher is better; downward arrow (↓): lower is better.

| Subject | Model | IncepV3 ↑ | CLIP ↑ | EffNet Dist. ↓ | SwAV Dist. ↓ | 50-way-top-1 Acc. ↑ ± std |
|---|---|---|---|---|---|---|
| Subj01 | BrainDiffuser | 0.914 | 0.923 | 0.710 | 0.407 | $0.53 \pm 0.23$ |
| | Int-BD | 0.851 | 0.885 | 0.782 | 0.470 | $0.46 \pm 0.23$ |
| | Int to CLIP | 0.942 | 0.955 | 0.655 | 0.412 | $0.69 \pm 0.20$ |
| Subj02 | BrainDiffuser | 0.909 | 0.907 | 0.725 | 0.417 | $0.52 \pm 0.23$ |
| | Int-BD | 0.842 | 0.869 | 0.798 | 0.480 | $0.42 \pm 0.23$ |
| | Int to CLIP | 0.946 | 0.952 | 0.661 | 0.412 | $0.70 \pm 0.20$ |
| Subj05 | BrainDiffuser | 0.917 | 0.928 | 0.702 | 0.409 | $0.55 \pm 0.22$ |
| | Int-BD | 0.860 | 0.886 | 0.771 | 0.466 | $0.47 \pm 0.23$ |
| | Int to CLIP | 0.941 | 0.951 | 0.668 | 0.417 | $0.68 \pm 0.20$ |
| Subj07 | BrainDiffuser | 0.886 | 0.900 | 0.736 | 0.430 | $0.50 \pm 0.21$ |
| | Int-BD | 0.831 | 0.868 | 0.801 | 0.484 | $0.43 \pm 0.23$ |
| | Int to CLIP | 0.933 | 0.950 | 0.668 | 0.416 | $0.68 \pm 0.20$ |

We also find that the specific distribution of voxel activations within the same region varies with each sub-concept examined. For instance, in Figure 4, the reference region for bodies (second group from the top) displays distinct activation patterns depending on the specific question. Even though all activations tend to overlap with the reference regions, the distribution of voxel intensity varies depending on the specific sub-concept. The question "Is someone running in the image?" results in a widespread pattern of low-level activations, interrupted by a few cluster of highly active voxels. In contrast, the question related to dancing yields a more intense activation of those same clusters, with little or no activation in the surrounding voxels. Similar comments can be made when comparing place-related activations (first group from the top in Figure 4), such as the questions "Is the image taken outdoors?", "Is the image taken on a street?". We don't observe a sharp distinction in the activations for faces and bodies. Neuroscientific studies have shown that face- and body-selective areas are anatomically close in the human (and monkey) high-level visual cortex and often co-activate when a whole person or animal is present [47]. This is simply explainable because, under normal circumstances, faces and bodies are seen together. For many of the questions related to bodies and faces we observed an overlap with both face and body regions. For example, the questions "Is the subject sitting?" and "Is the subject running?" show strong overlap with body-selective regions but also notable overlap with face areas.

## 4 Discussion

In this study we have presented an interpretable model for the decoding of visual stimuli from brain fMRI data. Prior work has largely explored how to obtain performance gains over state-of-the-art models; our study has taken a complementary approach. We believe that incremental advancement

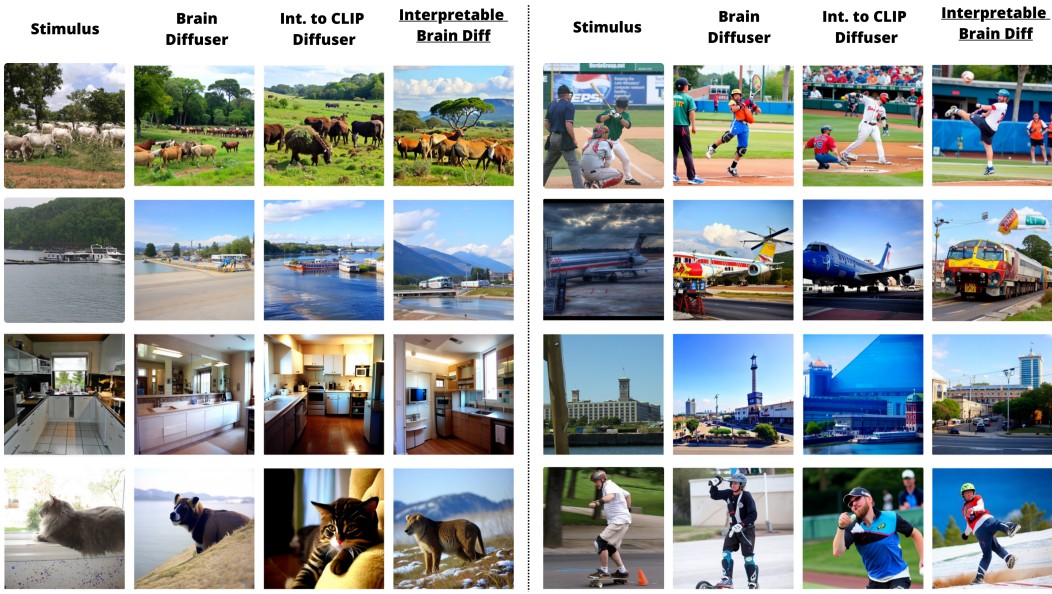

Figure 3: Comparison of our reconstruction results (columns 4, 8) to the original stimuli (columns 1, 5), the images reconstructed by the original BrainDiffuser (columns 2, 6) and those reconstructed by the Int to CLIP model (columns 3, 7). Results presented here refer to subj01.

is an incredibly valuable path of research. However, the field of brain decoding is starting to show signs of performance saturation. In this work, rather than contributing to the field solely with incremental performance gains, we focused on combining state-of-the-art decoding accuracy with clear, interpretable mechanisms, with the ultimate goal of unlocking deep insights into human brain neural functions. By mapping voxel signals to a human-readable semantic space before projecting them into CLIP embeddings, we reformulate decoding as a two-step problem, creating a bridge between low-level BOLD signals and human concepts. We have found that this semantic bottleneck allows for the extraction of rich neuroscientific insights with minimal performance loss, as the generated images incurred in only a minimal decline in both low- and high-level metrics.

Concept-specific voxel activations found in this study align well with canonical regions of the visual cortex, supporting their validity. Furthermore, the spread pattern of informative voxel activations reinforces the hypothesis of distributed rather than localized semantic encoding in the brain. As highlighted by the activation maps, the patterns of co-activations carry crucial information that is not completely expressed by the voxels location only.

To the best of our knowledge, this is the first complete brain decoding pipeline that introduces a truly interpretable intermediate representation. While post-hoc explainability methods - such as Grad-CAM [45], SHAP[33], or attention-based saliency - can technically be applied to existing models, these techniques generally rely on the assumption that the model's output space is inherently interpretable. In the context of brain decoding, this is often not the case: most models operate between two abstract spaces - neural activity and CLIP-like embeddings - making it challenging to extract meaningful insights from such explanations. This limitation motivated the design of our semantic bottleneck. By explicitly structuring the intermediate representation around human-interpretable concepts, our framework provides a more transparent link between brain activity and visual content. We believe this principled integration of interpretability into the core model architecture offers a more robust and actionable understanding than standard post-hoc techniques.

**Implications for neuroscience** Our work carries several implications for neuroscience. First, the semantic latent space offers a link from neural data to concepts, enabling voxel-level neuroscientific insights. Alignments with established ROIs support the validity of the decoding approach. Moreover, our findings support the hypothesis that, at fMRI scale, brain representations of visual concepts can be linearly approximated in semantic spaces similar to those learned by modern vision-language models,

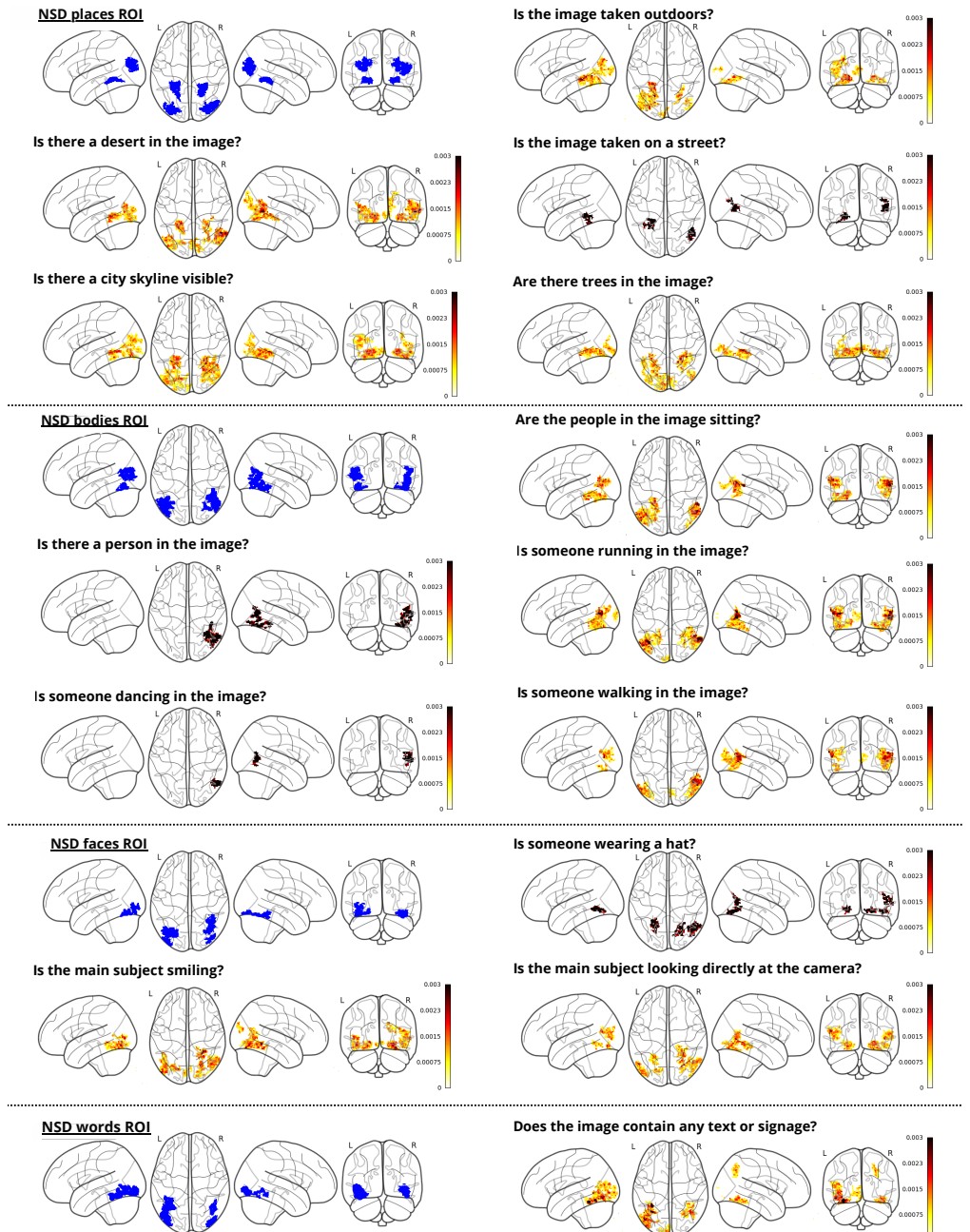

Figure 4: Activation maps projected in MNI-512 space, compared to category-specific reference maps. Sections correspond to places (top left), bodies (top right), faces (bottom left) and words (bottom right). Within each section, the blue map shown first presents the reference for that category. Other images show the activations derived from the interpretable matrix $A$. All maps are from subject01.

such as CLIP. Finally, the observed distribution of activation patterns underscores the complexity and redundancy of semantic representations in the brain.

**Model Interpretability vs. Performance Trade-off**    The introduction of a semantic bottleneck inevitably leads to a reduction in reconstruction quality. This is expected: the prompts are derived from human-authored COCO captions via BLIP-2 and GPT-4o, and human descriptions typically emphasize semantic content over geometric layout. Nevertheless, our curated question set does

include some geometry-oriented items. However, the measured drop in performance is small and offset by significant gains in interpretability and neuroscientific observations. The *Int to CLIP* model serves as an upper bound, revealing that the chosen interpretable space preserves most of the information required for faithful reconstructions.

**Methodological strengths**  The use of a large and high-quality dataset (NSD) ensures robust training and evaluation of the results obtained. The two-stage regression approach (brain → interpretable space → CLIP space) decouples decoding from semantic interpretation, allowing for model transparency. Importantly, the proposed framework is architecture-agnostic: the semantic bottleneck operates prior to the final CLIP embedding prediction, making it compatible with any decoding pipeline that maps brain activity to CLIP or similar multimodal embeddings, which is common among many decoding pipelines. Using GPT-4o and BLIP-2 for the generation of question-answer pairs makes the process of generating the interpretable space automated and scalable. Finally, the consistency of voxel activation patterns across different subjects confirms the robustness and generalizability of the method.

**Limitations**  Of course, this approach comes with some limitations. First, the introduction of the semantic bottleneck causes slight performance degradation in both low- and high-level reconstruction metrics compared to the non-interpretable BrainDiffuser baseline. Even though our main target is the semantic content of the reconstructed images some questions related to the picture geometry and the spatial positioning of the subjects are present in the chosen set. However, it is possible that these questions do not capture the whole spectrum of features present in complex NSD images, yielding decreased low-level metrics of reconstructed images. This leads to a second limitation of the approach: interpretability is constrained by the relevance and completeness of the generated question set, and the extension of interpretability for structural information is worth to be explored in the future. We also note that the overall quality of the pipeline is limited by the performance of the VQA model used to build the interpretable representations of the images, BLIP-2 in this case. Finally, our study is currently limited to the NSD dataset and to visual decoding tasks. Whether it is possible to generalize to other modalities remains to be demonstrated.

**Future directions**  Possible paths for future research include an expansion of the interpretable space to incorporate a wider and more diverse set of concepts, possibly hierarchical, in order to produce a more granular representation of the images, ultimately allowing for increased image reconstruction performance and, more importantly, a wider analysis of activation maps. The application of this framework to other datasets and modalities (such as auditory decoding or language processing) is possible, as well as its extension to future imagery data in which participants imagine visual scenes, objects, or concepts instead of viewing them directly, when these will be available. Finally, moving beyond pure semantics and integrating a greater set of structural interpretability elements (e.g. object layout, scene geometry, subjects position) could help in clarifying how the brain represents complex visual information.

**Privacy and Ethics considerations**  With the advancement of brain decoding and image reconstruction technologies, there is potential for misuse and privacy violations. Even though current models require the cooperation of the subjects and are strongly limited by the nature of the training data, future improvements may expand capabilities and it's important to ensure responsible use of brain data.

## 5   Conclusions

We investigated whether interpretability in visual brain decoding can coexist with high-fidelity image reconstruction. Our two-stage framework, initially mapping fMRI data into a semantic latent space and subsequently projecting into the CLIP latent space, quantitatively shows that this kind of model transparency imposes only a modest performance drop with respect to classical pipelines, while bringing the advantage of built-in interpretability. Moreover, the category-selective voxel patterns we found overlap with canonical hotspots on the visual cortex, providing neuroscientific validation and aligning with the idea that semantic content is distributed in the human brain. These results offer a practical way to study how concepts are represented in the brain.

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

## 6 Appendix

### 6.1 Evaluation of interpretable space $\mathcal{L}$

To determine whether the latent space $\mathcal{L}$ retains enough semantic detail to eventually be mapped reliably into the CLIP space by our Interpretable BrainDiffuser, we carried out the following analysis. For every image in the training and test partitions of NSD we extracted its CLIP-Vision embedding $\mathbf{c} \in$ CLIP and its interpretable embedding $\mathbf{l} \in \mathcal{L}$. We then fit a linear ridge-regression model $B : \mathcal{L} \to$ CLIP, selecting the regularization coefficient $\alpha$ on a logarithmic grid from $10^{-6}$ to $10^6$ by 5-fold cross-validation. After training, the model projected each test embedding $\mathbf{l}$ into a predicted CLIP vector $\hat{\mathbf{c}} = B\mathbf{l}$. Finally, we performed k-nearest-neighbor retrieval with $k \in \{1, 5, 10, 15\}$. For every test image we queried the original CLIP index with $\hat{\mathbf{c}}$ and recorded how often the ground-truth image appeared within the top-k returned results. The retrieval accuracy calculated in such way quantifies how faithfully information in $\mathcal{L}$ can be transferred to CLIP. Table 3 shows the accuracies obtained for the four values of $k$. These results confirms the quality of the designed interpretable representations, further supported by the fact that, even when the ground truth image is absent from the top-$k$ list, the nearest neighbors retrieved by the mapped embeddings typically show strong similarity with the query. Figure 5 displays a set of representative examples that illustrate this similarity.

Table 3: Top-$k$-Nearest-Neighbors retrieval accuracy achieved after projecting latent embeddings $\mathcal{L}$ into CLIP-Vision space and querying the original CLIP index.

| Metric | Accuracy |
|---|---|
| Top-1-NN | 29% |
| Top-5-NN | 59% |
| Top-10-NN | 74% |
| Top-15 | 82% |
| Chance level | 0.1% |

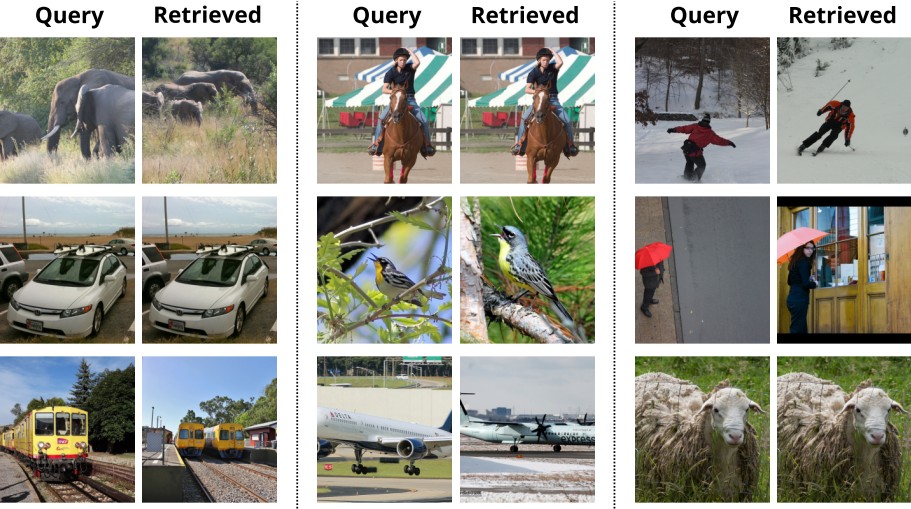

Figure 5: For 9 randomly selected test images (left in each pair), we show the nearest neighbor returned by 1-NN search in the original CLIP index after projecting the latent embedding $\mathcal{L}$ into CLIP space (right in each pair).

## 6.2 Reconstruction examples

Below, we present a series of images reconstructed by our interpretable model, along with their corresponding images generated by the original BrainDiffuser and the *Int to CLIP* model. These reconstructed images refer to subjects 2, 5 and 7, respectively. For the results relative to subject 1, please refer to the main section of the article under "Results."

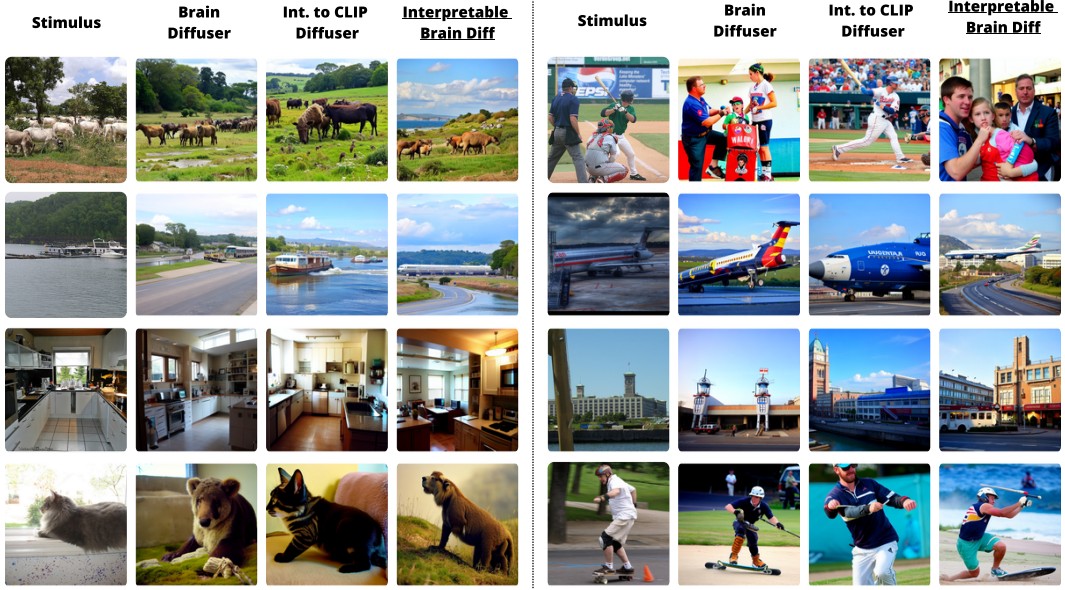

Figure 6: Comparison of our reconstruction results (columns 4, 8) to the original stimuli (columns 1, 5), the images reconstructed by the original BrainDiffuser (columns 2, 6) and those reconstructed by the Int to CLIP model (columns 3, 7). Results presented here refer to subj02.

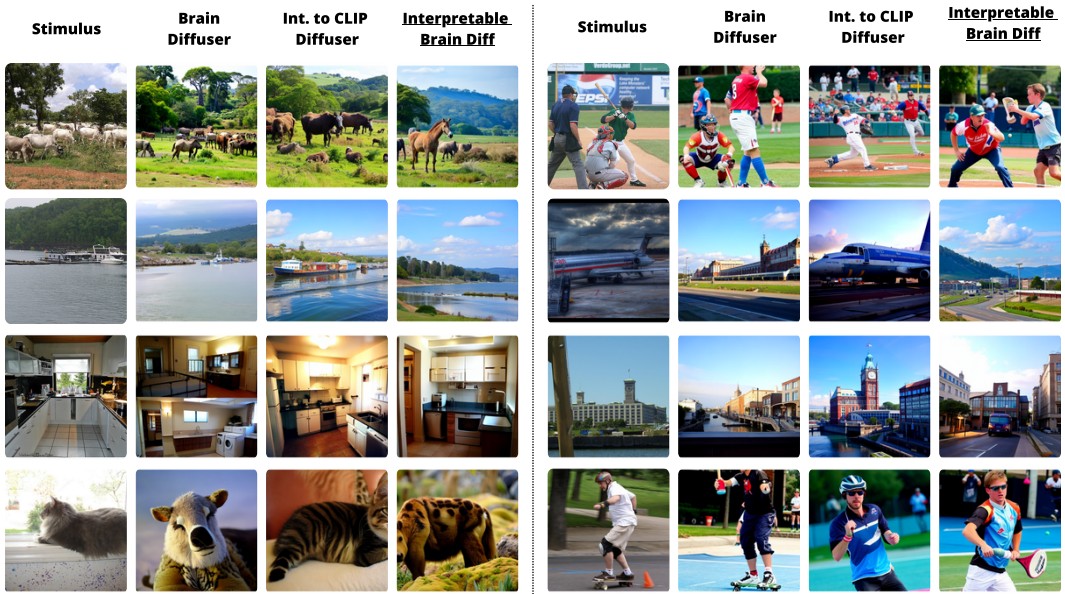

Figure 7: Comparison of our reconstruction results (columns 4, 8) to the original stimuli (columns 1, 5), the images reconstructed by the original BrainDiffuser (columns 2, 6) and those reconstructed by the Int to CLIP model (columns 3, 7). Results presented here refer to subj05.

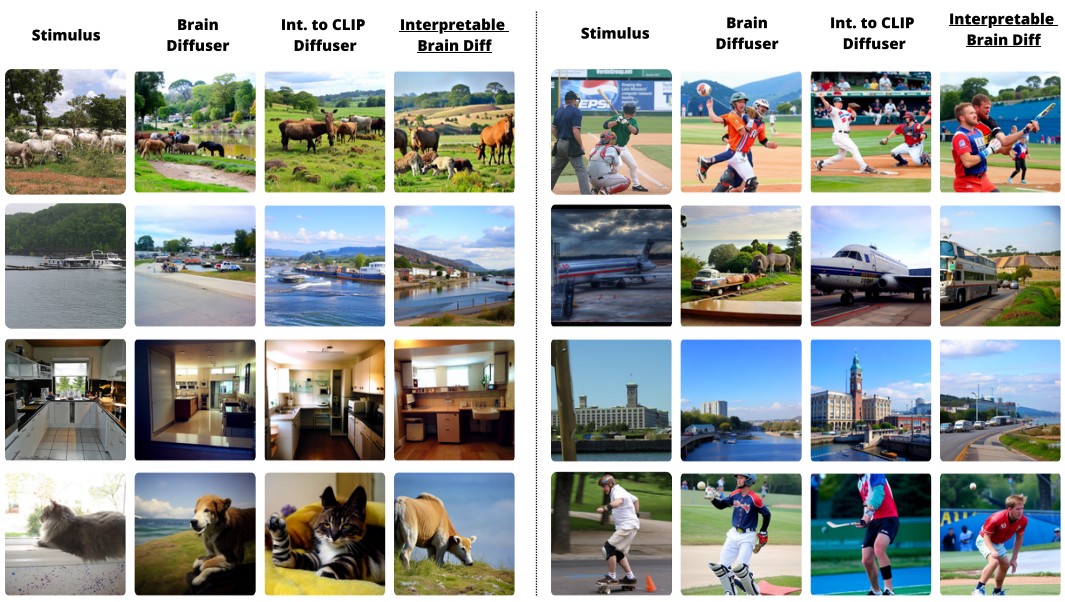

Figure 8: Comparison of our reconstruction results (columns 4, 8) to the original stimuli (columns 1, 5), the images reconstructed by the original BrainDiffuser (columns 2, 6) and those reconstructed by the Int to CLIP model (columns 3, 7). Results presented here refer to subj07.

## 6.3 Brain Regions Visualizations

In this section we present activation maps derived from the interpretable model for subjects 2, 5, and 7, visualized in MNI-512 space and compared against category-specific reference maps. For the corresponding figure related to subject 1, please see the "Results" section of this article.

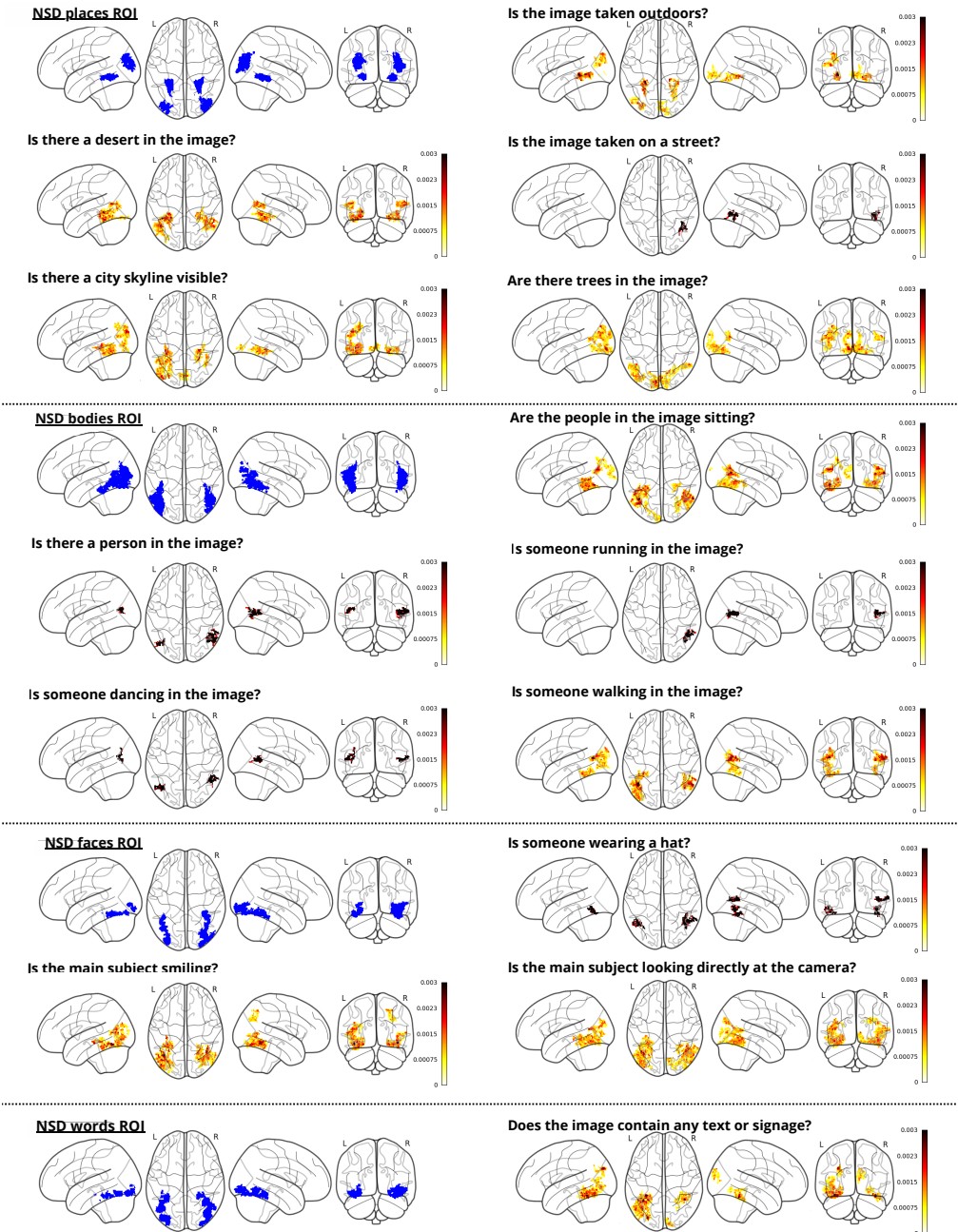

Figure 9: Activation maps for subject 2, projected in MNI-512 space and compared to category-specific reference maps. Sections correspond to places (top left), bodies (top right), faces (bottom left) and words (bottom right). Within each section, the blue map shown first presents the reference for that category. Other images show the activations derived from the interpretable matrix $A$.

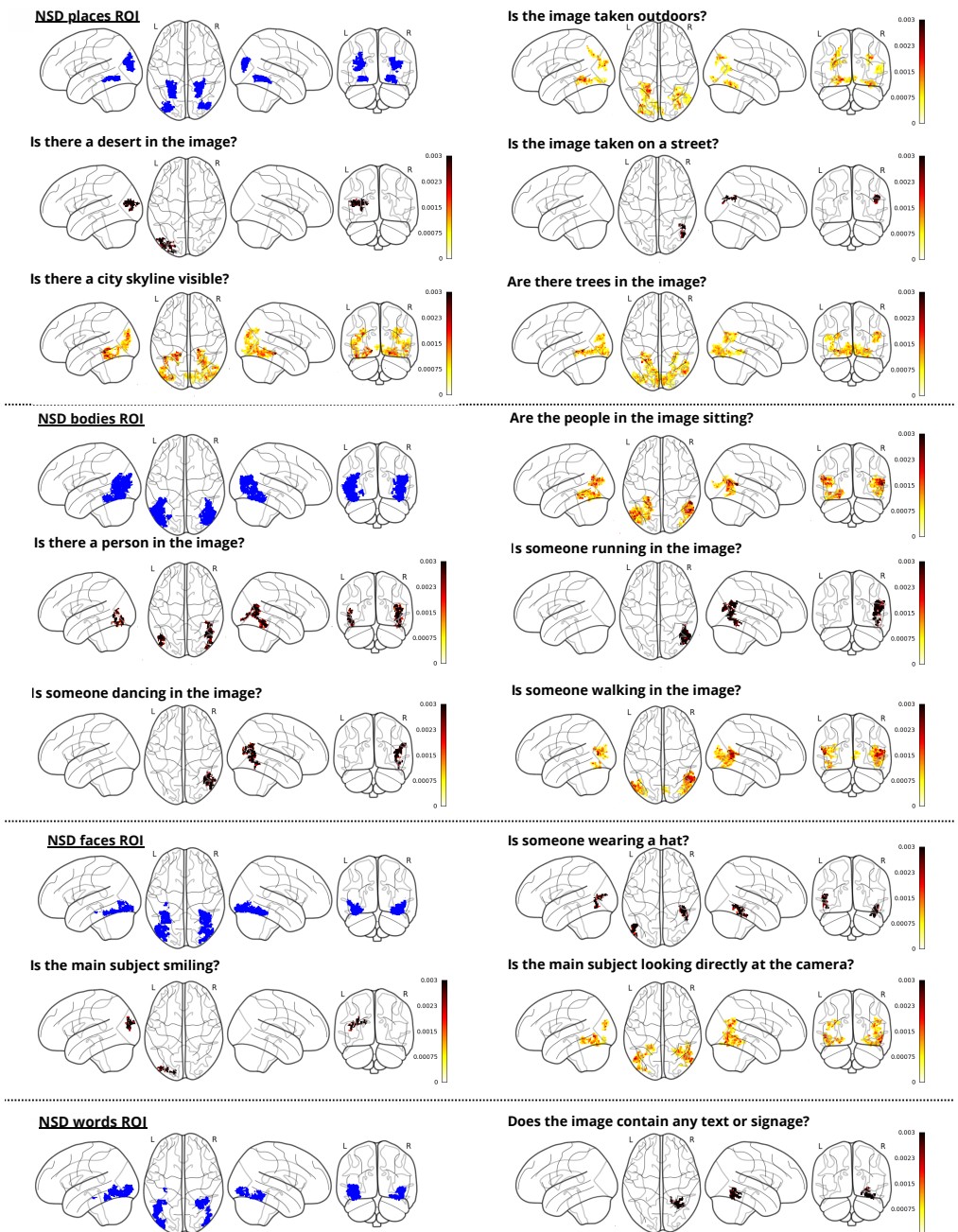

Figure 10: Activation maps, for subject 5, projected in MNI-512 space and compared to category-specific reference maps. Sections correspond to places (top left), bodies (top right), faces (bottom left) and words (bottom right). Within each section, the blue map shown first presents the reference for that category. Other images show the activations derived from the interpretable matrix $A$.

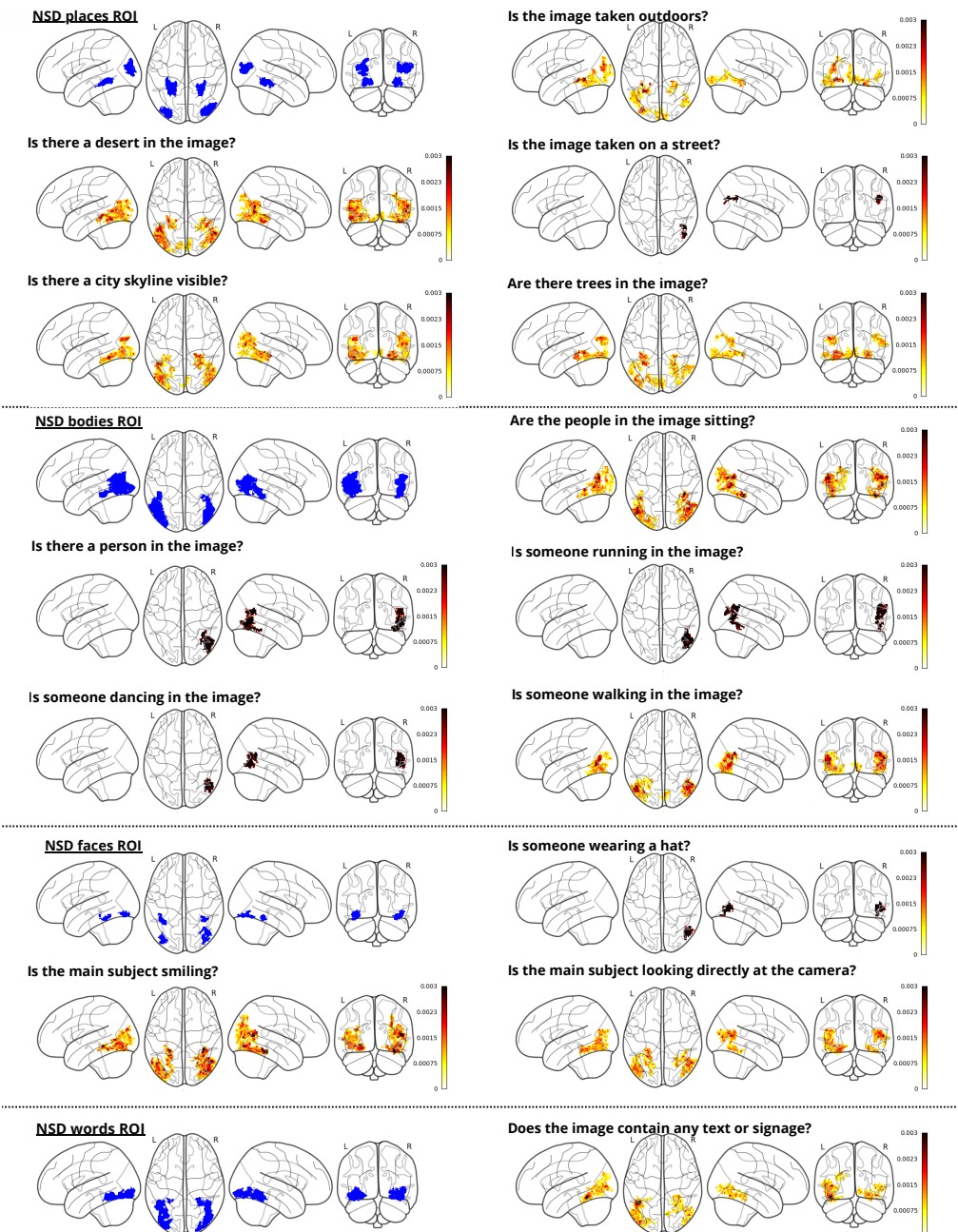

Figure 11: Activation maps for subject 7, projected in MNI-512 space and compared to category-specific reference maps. Sections correspond to places (top left), bodies (top right), faces (bottom left) and words (bottom right). Within each section, the blue map shown first presents the reference for that category. Other images show the activations derived from the interpretable matrix $A$.

### 6.4 Complete list of questions

1. Is there a person in the image?
2. Is there more than one subject?
3. Is the image taken indoors?
4. Is the image taken outdoors?
5. Is the image related to sports?
6. Is the image related to animals?
7. Is the image related to transportation?
8. Does the image contain water bodies like a sea, ocean, or river?
9. Are there any buildings visible in the image?
10. Are there trees in the image?
11. Is the image related to food?
12. Does the image contain any text or signage?
13. Is the image taken during the daytime?
14. Is the image taken at night?
15. Are vehicles present in the image?
16. Are there any children in the image?
17. Are the people in the image standing?
18. Are the people in the image sitting?
19. Is the image related to celebrations or parties?
20. Is the image related to a bathroom or restroom?
21. Is there a surfboard in the image?
22. Is someone riding a wave in the image?
23. Are people sitting on a rooftop?
24. Does the image show a toilet?
25. Are there zebras in the image?
26. Is there a shower visible in the image?
27. Are there animals standing on dirt ground?
28. Is there sand in the image?
29. Is there a mountain in the image?
30. Are there cars parked in the image?
31. Does the image include bicycles?
32. Are there birds in the image?
33. Are there plants or flowers visible?
34. Is someone holding an umbrella?
35. Is someone wearing a hat?
36. Is the image taken on a beach?
37. Are there boats in the image?
38. Is there a bridge in the image?
39. Are there stairs in the image?
40. Is there a window visible in the image?
41. Is there a table in the image?
42. Is there a chair in the image?
43. Is there a bed in the image?

44. Is there a lamp in the image?
45. Are there cups or glasses visible?
46. Are there plates visible?
47. Is there a computer or laptop in the image?
48. Is there a mobile phone visible?
49. Are there books or papers in the image?
50. Are there shelves in the image?
51. Is there a refrigerator in the image?
52. Is there a microwave in the image?
53. Is there a stove or oven visible?
54. Is there a washing machine in the image?
55. Are there curtains in the image?
56. Are there rugs or carpets visible?
57. Are mirrors present in the image?
58. Are there paintings or artworks on the walls?
59. Is there a clock visible in the image?
60. Is there a park in the image?
61. Is the image set in a forest?
62. Is there snow in the image?
63. Is it raining in the image?
64. Is there fog or mist in the image?
65. Are there mountains in the background?
66. Is there a desert in the image?
67. Is there a city skyline visible?
68. Is the image taken on a street?
69. Is there a marketplace in the image?
70. Are there shops or stalls visible?
71. Are there fences or barriers visible?
72. Is the image taken at an amusement park?
73. Are there swings or playground equipment visible?
74. Are there stairs or escalators in the image?
75. Is the image taken in a subway or train station?
76. Are there traffic lights in the image?
77. Is someone swimming in the image?
78. Is someone running in the image?
79. Is someone walking in the image?
80. Is someone cycling in the image?
81. Is someone playing a musical instrument?
82. Is someone reading in the image?
83. Is someone cooking in the image?
84. Is someone eating or drinking?
85. Is someone talking to another person?
86. Is someone taking a photo?
87. Is someone holding an object?

88. Is someone painting or drawing?

89. Is someone driving a vehicle?

90. Is someone fishing in the image?

91. Is someone dancing in the image?

92. Are people wearing jackets or coats?

93. Are people wearing swimsuits?

94. Are people wearing uniforms?

95. Are people wearing traditional clothing?

96. Are hats or caps visible?

97. Is the image symmetrical?

98. Does the image contain bright colors?

99. Are shadows visible in the image?

100. Are reflections visible in the image?

101. Is there smoke or fire in the image?

102. Does the image contain unusual patterns?

103. Are there pets in the image?

104. Are there insects visible in the image?

105. Does the image have signs of damage or destruction?

106. Are there fences or railings visible?

107. Is the main subject a human?

108. Is the main subject an animal?

109. Is the main subject a man?

110. Is the main subject a woman?

111. Is the main subject a child?

112. Is the main subject elderly?

113. Is the main subject a group of people?

114. Is the main subject alone?

115. Is the main subject smiling?

116. Is the main subject interacting with someone?

117. Is the main subject looking directly at the camera?

118. Is the main subject facing away from the camera?

119. Is the main subject partially visible (cropped)?

120. Is the main subject wearing formal clothing?

121. Is the main subject wearing casual clothing?

122. Is the main subject carrying an object?

123. Is the main subject holding a tool?

124. Is the main subject using electronic devices?

125. Is the main subject sitting on the ground?

126. Is the main subject climbing?

127. Is the animal a mammal?

128. Is the animal a bird?

129. Is the animal a reptile?

130. Is the animal an amphibian?

131. Is the animal a fish?

132. Is the animal a pet?
133. Is the animal a farm animal?
134. Is the animal a wild animal?
135. Is the animal flying?
136. Is the animal swimming?
137. Is the animal eating?
138. Is the animal drinking water?
139. Is the animal alone?
140. Are there multiple animals?
141. Are the animals interacting with each other?
142. Is the animal domesticated?
143. Is the main subject an object?
144. Is the object made of wood?
145. Is the object made of metal?
146. Is the object made of plastic?
147. Is the object broken?
148. Is the object old or vintage?
149. Is the object modern?
150. Is the object electronic?
151. Is the object artistic or decorative?
152. Is the object used for work or utility?
153. Is the main subject in the center of the image?
154. Is the main subject on the left side?
155. Is the main subject on the right side?
156. Is the main subject near the top?
157. Is the main subject near the bottom?
158. Is the main subject partially out of the frame?
159. Is there a background behind the main subject?
160. Is the main subject framed by other objects?
161. Is the main subject closer to the foreground?
162. Is the main subject farther in the background?
163. Does the image appear staged?
164. Does the image look candid or natural?
165. Is there movement captured in the image?
166. Is the image static?
167. Does the image look artistic or abstract?
168. Is the setting rural?
169. Is the setting urban?
170. Is the setting domestic?
171. Is the setting industrial?
172. Is the setting natural?
173. Is it sunny in the image?
174. Is it cloudy in the image?
175. Is there rain in the image?

176. Is it snowing in the image?
177. Are there visible shadows in the image?
178. Does the image depict sunset or sunrise?
179. Is the image taken during golden hour?
180. Is someone dancing?
181. Is someone playing sports?
182. Is someone cooking?
183. Is someone cleaning?
184. Is someone fixing or repairing something?
185. Is someone driving?
186. Is someone hiking?
187. Is someone fishing?
188. Is the image black and white?
189. Is the image edited or filtered?
190. Does the image use high contrast?
191. Is the image blurred?
192. Does the image have reflections?
193. Is the subject interacting with animals?
194. Is the subject interacting with machines?
195. Is the subject interacting with nature?
196. Is the subject interacting with others?
197. Is the subject interacting with water?
198. Is the subject wearing sunglasses?
199. Is the subject wearing a hat?
200. Is the subject wearing a uniform?
201. Is the subject wearing shoes?
202. Is the subject barefoot?
203. Is there a car visible?
204. Is there a bicycle visible?
205. Is there a bus visible?
206. Is there a train visible?
207. Is there an airplane visible?
208. Is there a boat visible?
209. Is there a motorcycle visible?
210. Is there symmetry in the image?
211. Is there repetition or patterns in the image?
212. Does the image depict destruction or ruins?
213. Are there visible tools or instruments?
214. Is there a stage or performance area visible?

