# OpenReview forum: "Bridging Brains and Concepts: Interpretable Visual Decoding from fMRI with Semantic Bottlenecks"
_NeurIPS.cc/2025/Conference — NeurIPS 2025 poster_

### Official Review · Reviewer_9695 · 2025-06-20

**Clarity:** 3
**Significance:** 3
**Originality:** 3
**Rating:** 4
**Confidence:** 3

**Summary:**

This paper proposes an interpretable fMRI-based visual decoding framework that integrates a semantic bottleneck into the established BrainDiffuser pipeline. The bottleneck consists of a 214-dimensional interpretable space, where each dimension corresponds to a yes/no question about the image (e.g., “Is there a person?”, “Is it outdoors?”). The authors demonstrate that the proposed model maintains competitive decoding accuracy. The voxel-wise contribution to semantic dimensions is visualized and aligns well with canonical category-selective regions in the visual cortex.

**Questions:**

1. Have the authors considered removing the reliance on pre-defined binary labels and instead allowing the model to directly generate human-readable explanations via a large language model (LLM)? Such an approach could potentially bypass the human bottleneck of manually crafting or filtering semantic dimensions, and allow for more flexible and context-aware interpretability.

2. From a machine learning perspective, the structure of a network optimized for interpretability may differ significantly from one optimized purely for performance. In this case, the original BrainDiffuser model is essentially being “reshaped” to fit the interpretability constraints—possibly at the cost of performance, as the observed degradation suggests. This raises a broader concern: are we truly interpreting how high-performing models behave, or are we just interpreting a specially trained, interpretable variant? Is there any potential approach to interpret existing high-performing models without retraining them specifically for interpretability?

3. Is it possible to train an image generator conditioned directly on the semantic bottleneck without using CLIP as an intermediate space? Could this allow a fully interpretable path: brain to concepts (bottleneck) to image?

**Ethical Concerns:**

["NO or VERY MINOR ethics concerns only"]

**Final Justification:**

The authors addressed most of my concerns. The paper is overall good. I maintain my original rating.

**Limitations:**

Yes

**Quality:**

3

**Strengths And Weaknesses:**

Strengths
1. Introducing a semantic bottleneck between fMRI and CLIP embeddings is a  feasible and concise way for interpretable decoding of brain activity. Although it cannot cover all perspectives, it can provide a certain degree of explanation at the semantic level that people are more concerned about.
2. From an experimental standpoint, the network maintains performance even with the added interpretability constraint, with only a minor drop in accuracy. Another commendable aspect is the inclusion of an upper-bound reference using ground-truth semantic labels (Int-to-CLIP). This model achieves better high-level semantic reconstruction, which is expected and provides a meaningful benchmark to evaluate the effectiveness of the proposed interpretable space. Beside that, authors also provide all the codes.
3. The authors provide brain region activation maps that support neuroscience validity. It can be seen that the voxel regions that support network semantic discrimination are consistent with the relevant regions considered in neuroscience.


Weaknesses
1. As the authors mentioned, the current interpretability relies on manually defined semantic information, specifically in the form of binary yes/no questions. This design inevitably omits many other potentially informative features, such as color, spatial layout, or fine-grained scene attributes. As a result, the scope of interpretation is limited, and the framework may miss more detailed or nuanced insights.
2. While the proposed methodology and research objective are indeed novel, the paper lacks a concrete articulation of the broader scientific significance. The current claims regarding the benefits of interpretability remain fairly general—for example, suggesting that the model "advances neuroscientific insight"—but these insights appear to align with already well-established findings in neuroscience. The added value of the proposed approach in uncovering new neural representations remains unclear.
3. The study compares the proposed model only against a single baseline (BrainDiffuser) with a relatively fixed architecture and training regime. This limited experimental design makes it difficult to assess the generalizability of the method. A broader comparison with alternative architectures or decoding paradigms would help strengthen the case for robustness and practical applicability.

---

> ### Author Rebuttal · Authors · 2025-07-30
>
> We thank the reviewer for the insightful comments.
>
> **Weakness 1:**
> We thank the reviewer for highlighting the limited scope of interpretable features in our semantic bottleneck. Our framework relies on binary questions automatically generated from COCO captions using BLIP-2 and GPT-4o. These captions focus mainly on object presence rather than spatial layout or fine-grained attributes, so our interpretable space reflects this bias.
>
> To ensure scalability across the large NSD dataset, we prioritized automated over manual generation. While effective for semantic coverage, this omits key perceptual dimensions. Extending the concept space to include structural and visual attributes could improve interpretability but requires more detailed language or structured templates not easily derived from captions. We will note this limitation and highlight future work to incorporate such features for more nuanced analyses of brain representations, especially in early visual areas.
>
> **Weakness 2:**
> We appreciate the reviewer’s concern regarding the articulation of broader scientific significance and the novelty of the neuroscientific insights derived from our model. Our primary aim in this work was to introduce a novel, interpretable framework for brain decoding to enable voxel-level understanding of semantic representations through an explicitly human-readable bottleneck.
>
> To validate this framework, we deliberately focused on a simple validation: testing whether the concept activation maps recovered by the model align with well-established functional specializations in the brain (face, place, body, and word-selective regions). That our model successfully recovers these canonical regions using linear mappings is not intended as a novel neuroscientific discovery per se, but rather as a validation of the framework’s interpretability and fidelity.
>
> We agree that the true potential of this approach lies in its ability to support more exploratory, fine-grained analyses such as identifying sub-regional distinctions within known ROIs or discovering previously overlooked patterns of representation. Indeed, our results in Figure 4 hint at this direction, as we observe meaningful variation in sub-concept activations (e.g., different bodily actions like “running” vs. “dancing” within the body-selective region), which may offer new insights into the distributed coding of semantic information in the cortex.
>
> To illustrate this potential we performed exploratory analyses inspired by the approach of the BrainDiffuser authors (section 3.6 in [1]) as well as BrainDiVE ([2]). Specifically, we aggregated activation maps across semantically related questions (e.g., all “face”-related questions) to obtain binary masks and used these to generate image reconstructions through the BrainDiffuser decoding pipeline. These reconstructions qualitatively reflected the expected semantic content (e.g., bodies or faces), indicating that the interpretable space preserves meaningful structure.
>
> Furthermore, we performed activation maximization analyses, identifying test images that elicited the strongest responses along specific semantic dimensions and asking three human raters to verify their relevance. The high correspondence between targeted concepts and top-ranked images ($67\%$) suggests that individual semantic dimensions are interpretable and consistent. While our aim here was to validate the approach rather than to make specific neuroscientific claims, these analyses demonstrate that this framework can, in principle, be used to probe cortical representations in new ways and potentially reveal novel insights.
> We will revise the manuscript to clarify this distinction between validating the approach and the future potential of the method for generating new hypotheses about brain representations.
>
> [1] Ozcelik, F. et al., Natural scene reconstruction from fMRI signals using generative latent diffusion. Sci Rep 13, 15666 (2023). https://doi.org/10.1038/s41598-023-42891-8
>
> [2] Luo, A. F. et al., 2023 Brain Diffusion for Visual Exploration: Cortical Discovery using Large Scale Generative Models. https://arxiv.org/abs/2306.03089
>
> **Weakness 3:**
> Thank you for this valuable point. We agree that the current comparison is limited to BrainDiffuser. We selected this model due to its strong performance, architectural clarity, and modular design, which allowed for a straightforward integration of our interpretable bottleneck. While a broader evaluation across multiple decoding architectures would indeed strengthen claims of generalizability, it was beyond the scope of this initial study.
>
> Importantly, the proposed framework is architecture-agnostic: the semantic bottleneck operates prior to the final CLIP embedding prediction, making it compatible with any decoding pipeline that maps brain activity to CLIP or similar multimodal embeddings (which is common among almost all the decoding pipelines). The decomposition into models $A$ and $B$ is modular by design, and can be easily incorporated into alternative or more complex architectures in future work.
>
> **Question 1:**
> We thank the reviewer for this interesting suggestion. The idea of leveraging a LLM to directly generate human-readable explanations from brain activity is indeed an intriguing and complementary direction. In some sense, this reverses our current approach: rather than constraining brain decoding through a predefined interpretable bottleneck, such a method would aim to decode free-form natural language directly.
>
> While prior work (references below) has explored LLM-based decoding of image captions or descriptive labels from neural data, such approaches tend to prioritize semantic fluency over interpretability. That is, while the outputs may appear meaningful or coherent, they often lack transparency in how specific neural features contribute to the generated explanation. In contrast, our framework explicitly exposes voxel-level contributions to discrete, interpretable concepts, making it possible to localize and visualize how different semantic dimensions are represented in the brain.
>
> That said, we agree this is a promising future direction. For example, fine-tuning an LLM to accept fMRI-derived embeddings - rather than image embeddings - as input could enable flexible, interactive decoding, where attention over fMRI embeddings could provide some form of interpretability. However, this would require fundamentally different modeling assumptions and training regimes, which we feel lies beyond the scope of the present work. We will include this discussion in the revised manuscript to highlight it as a compelling avenue for future research.
>
> A minimal set of works targeting the problem in this way:
>
> Qiu, W. et al., 2025 MindLLM: A subject-agnostic and versatile model for fMRI‑to‑text decoding. In Proceedings of the 41st International Conference on Machine Learning (ICML 2025) arXiv:2502.15786.
>
> MindLLM combines a neuroscience-informed encoder with a large language model to decode natural language from brain activity in a subject-agnostic way. Differently from our work, their output is a caption of the image derived from brain activity.
>
> Huang, W. (2024). BrainChat: Decoding semantic information from fMRI using vision–language pretrained models. arXiv:2406.07584
>
> BrainChat uses vision-language models like CoCa to generate image captions and answer questions directly from fMRI embeddings. Similarly, the output is a caption.
>
> Ye Z. et al., Generative language reconstruction from brain recordings, Nature Communications (2025). DOI:10.1038/s42003-025-07731-7.
>
> This study proposes a similar idea, but based on language, demonstrating how fMRI-derived embeddings can be fed into autoregressive LLMs to reconstruct continuous language, capturing fluent, context-aware text beyond candidate-based decoding.
>
> Luo AF et al., 2024 BrainDive: Uncovering and editing inner thoughts via fMRI-to-text decoding. arXiv:2312.00776
>
> BrainDive proposes a diffusion-based approach to generate images that maximize the predicted activity of specific brain regions, providing a form of inverse interpretability (you choose the region and get as output the image that is most likely to activate that region).
>
> Diffently from these and other work in the literature we target a specific problem: decoding images from brain activity in an interpretable way, by providing a minimal but effective modification to a robust decoding pipeline.
>
> **Question 2:**
> To the best of our knowledge, this is the first full brain decoding pipeline that incorporates a truly interpretable intermediate representation. While post-hoc explainability methods (e.g., Grad-CAM, SHAP, attention-based saliency) could, in principle, be applied to existing high-performing models, they typically assume an output space that is itself interpretable. In brain decoding, however, models often map between two inherently abstract spaces - brain activity and CLIP embeddings - making such explanations difficult to interpret meaningfully. This is precisely why we designed our framework around an explicit semantic bottleneck: to enable interpretable mappings at both the input and output levels, rather than relying on opaque latent dimensions.
>
> **Question 3:**
> Thank you for the interesting suggestion. In principle, it is indeed possible to fine-tune a diffusion model to condition directly on the semantic bottleneck, bypassing CLIP and enabling a fully interpretable end-to-end path from brain activity to concepts to images. However, in practice, this would require training the generator on a new set of conditioning vectors, and with the current dataset size, our previous experience in similar settings taught us that even efficient fine-tuning approaches like LoRA may not be sufficient to support high-quality image synthesis. For this reason, we considered such an extension beyond the scope of the present work, though we agree it represents an exciting direction for future research.

---

> > ### Comment · Reviewer_9695 · 2025-08-05
> >
> > I appreciate the authors' detailed responses and thorough discussion of the issues. After careful consideration, I believe the paper is overall good. While I find this an interesting piece of work, I still have some reservations about its practical applications. Therefore, I will maintain my original rating.

---

### Official Review · Reviewer_mU3X · 2025-07-02

**Clarity:** 3
**Significance:** 2
**Originality:** 2
**Rating:** 4
**Confidence:** 3

**Summary:**

This work proposes an interpretable framework to decode images from fMRI signals. They add a semantic bottleneck to an existing brain-decoding model (braindiffuser). This semantic bottleneck is basically a binary answer to 214 different questions related to images, which are generated by  LLM using the caption of each image. The authors evaluated the reconstruction performance of this model and also visualized interpretable activation maps. The reconstruction results don't get improved but the idea is claimed to be one step towards interpretable brain decoding models.

**Questions:**

* How did you choose the top 4% of most influential voxels? What do you define as the most influential voxels?
* Can you provide any more insights on your results in interpretability? Elaborate on why you think your method is actually capturing semantic information. Can you provide any quantitative metric for that?
* The motivation behind choosing Braindiffuser is not clear to me. Can you elaborate on that?
* In Figure 2, the ridge model (B) is not clear to me. Why do you have two B models? The first row seems to be confusing.(Also, aesthetics can get improved here)
* Can you compare your results regarding interpretability with any existing method?

**Ethical Concerns:**

["NO or VERY MINOR ethics concerns only"]

**Final Justification:**

The authors significantly addressed my concerns and questions, and I'm willing to raise my score from 2->4.

**Limitations:**

yes

**Quality:**

2

**Strengths And Weaknesses:**

**Strengths:**

* The paper is well-written and easy to follow.
* The proposed idea is interesting.
* It seems that a good amount of detail is provided on the method.

**Weaknesses:**

* The biggest flaw of the paper, in my opinion, is that although interpretability is the most important part of the paper, evaluations regarding that are rather limited and unconvincing. I would expect some sort of structured, summarized, and quantitative analysis on interpretable results besides the provided activation maps.  The provided activation maps for a subset of questions do not provide that much information. I have a hard time believing the method is actually interpretable without any aggregated quantitative results.
* The discussion on the results of interpretability is limited. Authors can provide better insight on that.
* The quality of schematic figures can be improved.
* There is no comparison with any other interpretable methods.

---

> ### Author Rebuttal · Authors · 2025-07-30
>
> We thank the reviewer for their comments and efforts towards improving our manuscript.
>
> **Weakness:**
> We thank the reviewer for highlighting this important point. We agree that interpretability is a central contribution of our work, and we acknowledge that the initial version of the manuscript emphasized qualitative results - particularly activation maps - over structured quantitative evaluations. This was motivated in part by the exploratory and conceptual nature of this work, where we aimed to demonstrate the feasibility of incorporating a semantic bottleneck into a full decoding pipeline. However, we fully agree that strengthening the quantitative analysis of interpretability would make the contribution more convincing.
>
> It is also worth noting that many interpretability methods in deep learning - such as attention visualizations, saliency maps, or Grad-CAM - are primarily evaluated qualitatively, often by inspecting whether visualizations align with expected input regions. Our work shares a similar spirit, but with a critical difference: our goal is not just to interpret a model’s output, but to interpret brain activity itself, through the lens of semantically meaningful intermediate representations. This makes interpretability not just a feature of the method, but the core objective.
>
> To better support our claims, we conducted a series of new analyses that will be included in the revised version:
>
> **ROI-based Dice Score Analysis:**
> We computed the Dice score between thresholded activation maps for concept-specific questions (e.g., body-related) and the corresponding functional ROIs (e.g., “bodies” ROI from NSD). Across multiple thresholds (50th, 75th, 80th, and 90th percentiles), we consistently observed that concepts like “body”, “face”, “place”, and “word” aligned more strongly with their expected ROIs than with unrelated ones, across all subjects.
>
> **ROI-wise activation decoding:**
> Inspired by the original BrainDiffuser analysis, we averaged activation maps across semantically grouped questions (e.g., all “face” questions), binarized them within NSD ROIs, and decoded them using the standard BrainDiffuser pipeline. The resulting reconstructions were qualitatively consistent with the original concept category, providing further evidence that the interpretable space captures meaningful structure.
>
> **Activation maximization analysis:**
> For each question, we identified test-set samples that produced the highest activation in the corresponding semantic dimension (via the trained Brain → Concept model). We asked three human raters to assess whether the associated images reflected the target concept. A success was recorded if at least 6 of the top 10 samples matched the intended concept. Across concepts and subjects, we observed a strong match rate ($67$%), supporting the idea that each dimension of the semantic space reliably corresponds to distinct visual categories.
>
> **Accuracy of answers in interpretable space:**
> We evaluated the accuracy of the predicted answers in the interpretable space by comparing them to ground-truth annotations derived from BLIP-2’s binary responses. This allowed us to assess the fidelity of semantic decoding independently of the final image reconstruction. (Please see our detailed response to Question 4 for results.)
>
> We will integrate these new analyses and expand the discussion accordingly in the final version of the paper, if accepted. We hope these additions address the reviewer’s concerns and help clarify the genuine interpretability enabled by our framework.
>
> **Questions:**
>
> 1) Thank you for the question. The decision to focus on the top $4\%$ of most influential voxels was made empirically, guided by two considerations. First, this threshold yields a voxel count that is comparable to that of well-characterized functional regions in the dataset, such as face-, body-, place-, and word-selective areas, each comprising a similar number of voxels ([2]). Second, this proportion provides a clear and interpretable visualization, which is particularly important given that our analysis is qualitative in nature. Importantly, our goal here is not to make strong quantitative claims about voxel importance but rather to offer intuitive insights into how different semantic concepts may be represented across the brain. Prior work in decoding and cognitive neuroscience suggests that while some high-level categories (e.g., faces, places) can be decoded from localized clusters, more fine-grained or abstract concepts often involve distributed representations (for example see ROI-wise experiment in Brain-Diffuser or BrainDive papers) and that’s why decoders benefit from large high dimensional voxel count in visual cortex. By visualizing the most influential voxels per concept, we aim to illustrate these patterns and highlight their correspondence with known functional regions. We will clarify this rationale more explicitly in the revised manuscript, if accepted.
>
> 2) We thank the reviewer for pointing out the need for greater clarity in the schematic figures. We agree that the original Figure 2 may not sufficiently differentiate between the distinct training phases of models A and B and their integration during inference. In response, we have redesigned this figure to more clearly illustrate the separation between the two training stages - (1) Brain → Concepts and (2) Concepts → CLIP - and the final inference pipeline. The revised schematic emphasizes that model $B$ is trained independently and frozen before being reused at inference time. Additionally, we improved Figure 1 to depict the full decoding pipeline more explicitly, making the overall flow from brain activity to image reconstruction easier to follow. Due to formatting constraints during the rebuttal phase, we are unable to include the updated figures here, but we will incorporate both revisions in the final version of the paper, pending acceptance.
>
> 3) Thank you for the question. We chose BrainDiffuser because of its strong performance, modular architecture, and simplicity of implementation, being basically a set of Ridge Regression between brain data and CLIP and VDVAE embeddings. Its clarity and flexibility made it well-suited for integrating our interpretable semantic bottleneck without introducing additional confounds. Our framework is architecture-agnostic: the semantic bottleneck is inserted before the CLIP embedding stage, which is common to most recent brain-to-image pipelines. We will clarify this rationale more explicitly in the revised manuscript. Another reason is that it was recently demonstrated that decoders based on linear architectures better generalize across different tasks, for example being able to decode imagery vision tasks [1] when trained on the same subjects, generalizing better than more complex and non-linear architectures.
>
> 4)  To assess the quality of the predictions in the interpretable concept space, we compared the model’s outputs against binary labels derived from BLIP-2 responses on the same set of images. Specifically, we thresholded the continuous outputs of the Brain → Concept model and computed several standard classification metrics across all $214$ questions. Our results are as follows:  Mean Label Accuracy: $0.83$, F1 Score (Micro): $0.78$, F1 Score (Macro): $0.47$, ROC-AUC (Micro): $0.91$, ROC-AUC (Macro):  $0.77$. These results indicate that the model is able to accurately decode high-level semantic attributes from brain activity, achieving strong overall performance in both classification accuracy and discriminability. The lower macro F1 score reflects the varying difficulty across concepts, as some binary questions are inherently more ambiguous or less visually salient. Nevertheless, this evaluation confirms that the interpretable space reliably captures meaningful semantic structure. We will include these results and clarify their implications in the revised manuscript, if accepted.
>
> 5) To the best of our knowledge, this is the first complete brain decoding pipeline that introduces a truly interpretable intermediate representation. While post-hoc explainability methods - such as Grad-CAM, SHAP, or attention-based saliency - can technically be applied to existing models, these techniques generally rely on the assumption that the model’s output space is inherently interpretable. In the context of brain decoding, this is often not the case: most models operate between two abstract spaces - neural activity and CLIP-like embeddings - making it challenging to extract meaningful insights from such explanations. This limitation motivated the design of our semantic bottleneck. By explicitly structuring the intermediate representation around human-interpretable concepts, our framework provides a more transparent link between brain activity and visual content. We believe this principled integration of interpretability into the core model architecture offers a more robust and actionable understanding than standard post-hoc techniques, and we will highlight this more clearly in the revised manuscript.
>
> It's important to note that the scientific discourse around interpretability in brain decoding models is still developing and our work aims to contribute to this conversation by proposing a simple and human-readable approach. While we acknowledge the limitations and are working to address them through additional analyses, we would kindly ask the reviewer: are there specific improvements or additions that, in your view, would make this work sufficiently valuable to be part of the ongoing scientific dialogue?
> We see this paper as a starting point rather than a final answer, and we would be grateful for your perspective on what would elevate the contribution further.
>
> [1] Kneeland R et al., 2025. NSD-Imagery: A benchmark dataset for extending fMRI vision decoding methods to mental imagery.
> [2] Allen, E.J. et al., 2022 A massive 7T fMRI dataset to bridge cognitive neuroscience and artificial intelligence.

---

> > ### Comment · Reviewer_mU3X · 2025-08-03
> >
> > I thank the authors for addressing my concerns and questions significantly. I raised my score from 2 -> 4.

---

### Official Review · Reviewer_zVsQ · 2025-07-03

**Clarity:** 3
**Significance:** 4
**Originality:** 3
**Rating:** 5
**Confidence:** 4

**Summary:**

This paper introduces a novel framework for interpretable visual decoding from fMRI data. The central idea is to address the "black box" nature of high-performing brain decoding models by inserting a human-readable "semantic bottleneck" into the existing BrainDiffuser pipeline. The authors factor the complex mapping from fMRI voxels to CLIP embeddings into two simpler, linear stages: a first mapping from brain activity to a 214-dimensional interpretable concept space, and a second mapping from this concept space to the CLIP embeddings. The interpretable space is constructed from a set of binary questions about image content (e.g., "Is there a person?"). The key contribution is that the first linear mapping provides direct, voxel-level insights into how semantic concepts are represented in the brain, with only a minor trade-off in final image reconstruction quality.

**Questions:**

- To improve clarity, would the authors consider revising Figure 2 to explicitly separate the training procedures for models A and B from the final, end-to-end inference pipeline? This would help readers better understand the elegant two-stage data flow.
- The current framework seems to require retraining the Concepts -> CLIP model (B) from scratch if the interpretable space is expanded. Have the authors considered more efficient update strategies, such as fine-tuning or modular approaches, to add new concepts without complete retraining?
- The original BrainDiffuser uses a structural stream to generate an "initial guess." It's not entirely clear from the text how this structural component interacts with the new interpretable semantic stream. Does the Int Brain Diff model still use the VDVAE-based initial guess, or does it rely solely on the CLIP embeddings derived from the interpretable space? Clarifying this would help in understanding the source of the performance drop in low-level metrics.
- The interpretable space was set to 214 dimensions based on "manual inspection" (Section 2, "Interpretable space"). Could the authors elaborate on the criteria used for this selection and whether they explored how varying the number of concepts affects the trade-off between reconstruction quality and the quality of the neuroscientific insights?

**Ethical Concerns:**

["NO or VERY MINOR ethics concerns only"]

**Limitations:**

yes

**Quality:**

3

**Strengths And Weaknesses:**

**Strengths**

**Elegant Solution to a Key Problem:** The paper addresses the critical trade-off between performance and interpretability in brain decoding. The proposed semantic bottleneck is a simple yet powerful method for gaining neuroscientific insights without a major loss in decoding accuracy.

**Novel Method for Creating Interpretable Space:** The use of LLMs (GPT-4o and BLIP-2) to automatically generate a human-readable semantic space is a scalable and innovative approach that moves beyond hand-crafted features (Section 2, "Interpretable space").
Strong Neuroscientific Validation: The primary claim of interpretability is well-supported by the results in Figure 4. The alignment of the learned concept maps with canonical Regions of Interest (faces, places, bodies, words) provides compelling evidence that the model is capturing meaningful brain representations.

**Thorough Quantitative Evaluation:** The authors provide a comprehensive comparison against the BrainDiffuser baseline using a wide range of low-level and high-level metrics (Tables 1 and 2). The inclusion of the "Int to CLIP" control model is a clever experimental design choice that effectively demonstrates the semantic richness of the interpretable space itself.
Reveals Deeper Brain Representation Patterns: The finding that activation patterns within a region vary for sub-concepts (e.g., running vs. dancing in the 'bodies' ROI, Figure 4) offers a nuanced insight into how the brain encodes information, supporting the theory of distributed representations.

**Weaknesses and Possible Improvements**

**Clarity of the Model Diagram (Figure 2):** The manuscript would benefit significantly from a revision of Figure 2. The current figure ambiguously merges the training process for the Concepts -> CLIP model (B) with the final inference pipeline. This makes it difficult to understand that the top and bottom rows represent distinct processes, how models A and B are trained with their respective targets, and that model B is frozen and reused. A clearer diagram separating these stages would greatly improve the paper's accessibility.

**Scalability of the Interpretable Space:** The current framework requires retraining the Concepts -> CLIP model (B) from scratch if the set of questions is expanded. This is a practical limitation for future research aiming to build on this work with more granular concepts. The discussion could be strengthened by acknowledging this and suggesting potential avenues for more efficient updates.

**Limited Discussion on Structural Information:** The paper notes that low-level reconstruction metrics see a larger performance drop (Table 1) and that the question set might not capture the "whole spectrum of features." The work is heavily focused on semantic concepts. An improvement would be to discuss more concretely how structural concepts (e.g., object layout, scene geometry)—a key component of the original BrainDiffuser's success via its VDVAE stream—could be integrated into this interpretable framework.

---

> ### Author Rebuttal · Authors · 2025-07-30
>
> We thank the reviewer for their comments. Below, we address the points raised under Weaknesses and provide point-by-point responses to the reviewer’s questions.
>
> **Clarity of Figure 2**
> We agree that the current version of Figure 2 may be unclear about the distinct training procedures for models A and B with the final inference pipeline. We appreciate the reviewer highlighting this concern. In response, we have designed a revised version of the figure that clearly separates the training stages (for the Brain → Concepts and Concepts → CLIP mappings) from the final end-to-end decoding process. The updated figure segregates (i) Brain → Concepts training, (ii) Concepts → CLIP alignment, and (iii) inference—using colour‑coded backgrounds so the ‘two B blocks’ can no longer be confused. The new figure disentangles the role of each component and clarifies that model B is frozen after training and reused in inference. While we are unable to include the revised figure during the rebuttal phase due to formatting constraints, we will incorporate this improvement in the final version of the paper, pending acceptance.
>
> **Scalability of the Interpretable Space**
> We appreciate this important observation regarding the scalability and extensibility of the semantic concept space. Currently, the Concepts → CLIP mapping (model B) must be retrained from scratch if the dimensionality of the concept space changes. This is a limitation of the current design, which we will explicitly discuss in the revised manuscript. (Please ssee reply to question 2 below)
>
> **Limited Representation of Structural Information**
> We agree that our current interpretable space is predominantly semantically driven and lacks an adequate representation of structural or spatial features, which are critical for low-level image fidelity. As the reviewer correctly points out and as we touched upon in the discussion this likely contributes to the greater performance drop in low-level reconstruction metrics compared to high-level semantic alignment.
> This limitation is mainly due to two factors: (i) our question-generation pipeline relies on LLM-based summarization of human annotators captions; these tend to emphasize semantic content over spatial layout or geometry and (ii) formulating fine-grained structural questions in a generalizable and scalable way is nontrivial, especially with natural images that lack consistent object placement (see reply 2 and 3). We will include these comments in the discussion of the revised manuscript, if accepted.
>
> **Questions:**
>
> 1) As noted above, we have revised Figure 2 to separate training and inference stages, clearly indicating the training flow for models A and B and the reuse of model B during inference.
>
> 2) We thank the reviewer for raising this important point regarding the extensibility of the interpretable concept space. In the current formulation, expanding or modifying the set of questions (semantic concepts) requires retraining the Concepts → CLIP mapping (model B) due to changes in input dimensionality. This presents a practical limitation for researchers who may wish to refine or scale the semantic bottleneck for different experimental goals or datasets.
> To this end, our framework was intentionally designed to be simple and modular, with both mappings - A and B - implemented as linear ridge regressions. This allows for interpretability and analytical tractability but currently lacks built-in flexibility for dynamic updates to the concept space.
> Nonetheless, we believe the design does admit potential extensions that could make it more scalable:
> - For applications where only a subset of concepts is needed (e.g., to test minimal sets for decoding or neuroscientific interpretability), model $A$ can be trivially subsetted by selecting the corresponding rows of its output. However, model $B$ would need to be retrained to project from this reduced concept space to the CLIP embedding space.
> - On the other hand, additional questions could be introduced by training a new regressor (model $A_2$) to predict only the added concept dimensions from brain activity. The outputs of model $A$ and $A_2$ could then be concatenated to form an augmented brain-to-concept representation (model $A^*$). However, model B must still be retrained to accommodate the new input dimensionality. In principle, this retraining could be implemented as fine-tuning (rather than from scratch), assuming the new concepts form a small perturbation of the original representation.
> - Another avenue for future work could be  to explore the pretraining model $B$ on a much larger, overcomplete set of interpretable concepts and later sparsifying it for specific use cases.
> We will elaborate on these ideas in the revised manuscript to better inform future work on the composability and scalability of interpretable decoders.
>
> 3) We appreciate the reviewer’s observation on the limited representation of structural information in our interpretable concept space. This limitation is acknowledged and contributes to the reduced performance on low-level reconstruction metrics (e.g., MSE, SSIM) compared to high-level semantic alignment. Although our framework retains the structural stream from the original BrainDiffuser (based on VDVAE), the semantic bottleneck remains biased toward high-level content.
> The bias toward semantic content in our interpretable space stems from its construction: questions are automatically generated from human-written COCO captions using BLIP-2 and GPT-4o, which typically highlight object presence rather than spatial arrangement. Natural language favors concise descriptions of what is in the image over how elements are positioned, making structural features underrepresented. Additionally, questions related to structural features of the image are harder to formulate at scale, as they require more complex, context-dependent descriptions of object relationships, which are rarely found in the source captions.
> In future work, we aim to address this limitation by enriching the dataset with spatially detailed captions (e.g., via re-captioning with multimodal LLMs) and regenerating the question set, or by explicitly designing structural questions. While our main goal was to demonstrate the feasibility of a semantic bottleneck for interpretable brain decoding, we agree that incorporating structural information remains a key direction for future research.
>
> 4) About the question regarding the rationale behind the choice of $214$ dimensions for the interpretable space: the decision was guided by a combination of practical and conceptual considerations. First, from the perspective of prior literature in cognitive neuroscience and language-based encoding models [1], relatively low-dimensional semantic feature spaces - often ranging from 30 to 100 dimensions - have been shown to capture a substantial portion of the explainable variance in neural responses. These findings suggested that high decoding performance does not necessarily require an extremely high-dimensional representation, especially when the features are carefully chosen for their informativeness and relevance.
> Second, follow-ups [2] the original BrainDiffuser framework demonstrated strong performance using low-rank approximations of the CLIP embedding space, often in the range of 100 to 200 dimensions. This indicated that a large fraction of the meaningful visual-semantic information could be retained even after significant dimensionality reduction. We therefore hypothesized that a semantically curated bottleneck in a similar dimensional range would be sufficient to preserve downstream decoding quality, while offering the added benefit of interpretability.
> Third, and perhaps most importantly, our decision was informed by manual inspection of the candidate questions generated through our automated BLIP-2 + GPT-4o pipeline. After running the question-generation pipeline on the image captions, we selected a set of $214$ binary questions that appeared semantically diverse and representative of common image content categories. This process was not performance-optimized; our goal was to create a broad, human-readable semantic space to test the framework’s feasibility in a realistic, scalable setting with automated rather than hand-tuned concept generation. We agree with the reviewer that exploring the effect of concept space dimensionality on both reconstruction quality and neuroscientific insight is a rich direction for future research. For instance, one might ask: how many questions are necessary to achieve a particular level of decoding performance? Is there a minimal concept set that still enables interpretable decoding above chance? Conversely, how much performance can be recovered by progressively increasing the number of concepts, and is it possible to match or even exceed the vanilla BrainDiffuser performance using only interpretable representations?
> While we are eager to investigate these questions, doing so at scale was beyond the scope of the current study due to computational constraints. Each configuration requires regenerating question-answer pairs using BLIP-2 across the full training set- a process that currently takes several days on several high-end H100 GPUs - and retraining both $A$ and $B$ mappings. We will discuss these limitations and opportunities in the revised manuscript to better contextualize the current design choices and motivate future extensions of the framework.
>
>
>
> [1] Benara V, Singh C, Morris JX, Antonello RJ, Stoica I, Huth AG, Gao J. Crafting Interpretable Embeddings for Language Neuroscience by Asking LLMs Questions. Adv Neural Inf Process Syst. 2024;37:124137-124162. PMID: 40276238; PMCID: PMC12021422.
>
> [2] Mayo, D., Wang, C., Harbin, A., Alabdulkareem, A., Shaw, A., Katz, B., & Barbu, A. (2024). BrainBits: How Much of the Brain are Generative Reconstruction Methods Using? In Advances in Neural Information Processing Systems 37 (NeurIPS 2024), Main Conference Track.

---

### Official Review · Reviewer_YAzb · 2025-07-03

**Clarity:** 3
**Significance:** 3
**Originality:** 4
**Rating:** 5
**Confidence:** 5

**Summary:**

This paper presents a semantically-interpretable fMRI-> image decoder model by mapping the brain responses to a set of 214 binary questions (such as “Is someone swimming in the image?”). This not-so-latent 214-D space is then used as the target for a linear regression from the fMRI regions to the binary vector describing the image. In addition, a linear mapping from this space to CLIP embeddings is trained (using a pre-trained Brain Diffuser model), so that the brain-to-image mapping is via this human-interpretable space. This bridges the gap between latent spaces and human concepts, making the interpretation of brain activations more semantic. In particular, they show that the linear regression weights from relevant questions map onto brain regions known to represent bodies, faces, places, and words.

Finally, they show that the reconstruction quality of the images is only somewhat reduced from using a more opaque latent space.

**Questions:**

Did you do any manual evaluation of the performance of BLIP on your questions? It seems like you should select a random sample of question-answer pairs to estimate the performance, as this could impact your results.

Why didn’t you use logistic regression to predict the semantic features? You’re predicting a binary variable, so it seems like logistic regression is much more appropriate.

Some of the questions presuppose a person in the image (such as questions 34-35, 77-94, 180-187, 193-202, etc.), some more than others (e.g., is the subject barefoot). Did these also correspond to face and/or body areas?

Why did you only look at one question with respect to face areas?

**Ethical Concerns:**

["NO or VERY MINOR ethics concerns only"]

**Final Justification:**

This is an exciting paper, and gives very good results. I have read the authors' response to my review, and they have clarified all of my concerns.

**Limitations:**

Yes.

**Quality:**

4

**Strengths And Weaknesses:**

Quality
+ the quality of the reconstructed images is still quite good compared to the original Brain Diffuser results.
+ The idea of using a semantic space is an interesting innovation.

Clarity

+ The model is well-described.
- It was unclear to me exactly how the questions and answers were constructed using GPT-o. What was the prompt used?
- can you explain what the 2-way accuracy in Alexnet latent space refers to? Similarly, 2-way and 50-way in the following sentences.

There are a bunch of places where the English could be improved.
Everywhere in the ms: “performances” should be replaced by “performance”. There are a few specific ones that need further adjustment, listed below.
line 52: dataset . -> dataset.
line 60: wouldn’t allow -> doesn’t allow
line 69: performances -> performance (and everywhere else this word is used).
line 85: performances -> performance by
line 93: a LLM -> an LLM
line 136: to condition -> conditioning
line 165: with in mind the idea -> with the idea in mind
line 166: reduce -> reduces
line 167: from previous -> from a previous
line 203: models -> model
lines 204-205: “The extraction of interpretable latent embeddings with BLIP2 took approximately 24 hours per subject” Why per subject? Didn’t all subjects see the same images?
line 214: reconstructing performances -> reconstruction performance
line 216-217: reconstructing accuracies -> reconstruction accuracy.
line 297: the training data nature -> the nature of the training data
line 304-305:
Moreover, the found category-selective voxel patterns overlap with canonical hotspots
->
Moreover, the category-selective voxel patterns we found overlap with canonical hotspots

Significance

+ This is a nice way to extract semantics from brain images. But have you compared this to the stuff Tom Mitchell did back in the day of training models to answer simply semantic questions from brain images? There might be better questions in his work.

Originality

+ I don’t know (besides Tom Mitchell’s work) of anyone else who has done this.

---

> ### Author Rebuttal · Authors · 2025-07-30
>
> We thank the reviewer for their valuable comments and insights. We proceed to clarify the points raised in the following and in the revised manuscript.
>
> **Question/answer pair generation:**
> We thank the reviewer for pointing this out and appreciate the opportunity to clarify.
> For each subject in the Natural Scenes Dataset (NSD), we aimed to generate a compact, interpretable representation of each image. To do so, we first extracted the first COCO caption associated with each image in the NSD training and test set. As a reminder, these captions were originally created by human annotators and are part of the COCO dataset, which underlies NSD.
> We collected all these captions and submitted them to GPT-4o with the following prompt:
> “Given the following set of captions, each representing an image, generate a set of 256 binary questions that are suitable to well describe the content of the images. Generate questions that are diverse and non-overlapping, and that describe as completely as possible all the images. These should maximize your information about the image content.”
> The set generated by GPT was not perfect; we manually inspected all questions to remove duplicates and semantically overlapping entries. This resulted in a final set of $214$ unique binary questions.
> These questions were then posed sequentially to the BLIP-2 vision-language model for each image in the dataset with the following question-specific prompt: “Question: [one question, for example ‘is the subject smiling?’] Answer:”.
> For every image, the model was queried 214 times, once per question, producing a binary (“Yes”/“No”) answer for each. The answers were converted to binary values (1 for “Yes”, 0 for “No”), resulting in a 214-dimensional interpretable vector representation for each image.
> This procedure was repeated independently for each subject in NSD, because in the original experiment images of the training sets were different across subjects, whereas the test images were shared by all subjects.
>
> **Clarifications on the used metrics:**
> Regarding the 2‑way and 50‑way accuracy metrics: these metrics evaluate how well reconstructed images preserve the semantic content of the original images using ViT, a pretrained Vision Transformer, as a stand-in for human perception. For each original image, the model predicts its most likely class (serving as a pseudo ground truth). We then classify the reconstructed version of the same image with the same model to obtain its class probabilities. Instead of evaluating accuracy over all ImageNet classes, we create smaller, simulated classification tasks. In one case, we compare the true class against just one randomly chosen incorrect class (2‑way accuracy); in another, we compare it against 49 randomly chosen incorrect classes (50‑way accuracy). These tasks are repeated 1000 times with different randomly chosen distractor classes to estimate the mean and variability of the accuracy for each reconstructed image. Averaging across all images produces an overall measure of how much class‑level semantic information is preserved in the reconstructions.
> AlexNet(2) and AlexNet(5) report the same 2‑way identification protocol but carried out in feature space rather than label space. We extract activations from layer 2 or layer 5 of a pre‑trained AlexNet and evaluate whether the reconstruction is closer to its true partner than to a single randomly chosen distractor image. Repeating the distractor draw for all possible images provides a robust estimate of identification accuracy at two levels of visual abstraction: layer 2 emphasises low‑level edges/textures, whereas layer 5 captures object‑level semantics.
>
> **Questions:**
>
> **Did you do any manual evaluation of the performance of BLIP on your questions?**
> We thank the reviewer for this very interesting and constructive suggestion. We agree that manually evaluating the performance of BLIP on our question set would provide valuable insights, as the final results are indeed influenced by the choice of the underlying VQA model. While we have not performed such an analysis yet, we plan on including it in the revised manuscript, if accepted, for a subset of question–answer pairs (5\% of the test set images selected randomly) to provide the readers with an estimate of its accuracy. Furthermore, we will explicitly acknowledge in the limitations section that the quality of the interpretable space is impacted by the VQA model. Improving the quality of the interpretable space, both from the question design point of view and on the VQA model used, is an important future direction.
>
> **Why didn’t you use logistic regression to predict the semantic features?**
> We agree that logistic regression seems like a natural choice for this type of problem; in the initial stages of this study we indeed experimented with predicting semantic features using logistic regression. However, when we compared its performance with that of Ridge regression trained to predict the standardized interpretable vectors directly, we found that the Ridge approach gave better results. Specifically, after thresholding, the linear regression model achieved a mean label accuracy of $0.83$, an F1 micro score of $0.78$, and a macro F1 score of $0.47$, with a Hamming loss of $0.17$. Its ROC-AUC was $0.91$ (micro) and $0.77$ (macro). In contrast, logistic regression performed worse, with a mean label accuracy of $0.63$, an F1 micro score of $0.56$, and a macro F1 score of $0.49$, along with a higher Hamming loss of $0.37$ and lower ROC-AUC values of $0.68$ (micro) and $0.71$ (macro).
>
> **Some of the questions presuppose a person in the image (such as questions 34-35, 77-94, 180-187, 193-202, etc.), some more than others (e.g., is the subject barefoot). Did these also correspond to face and/or body areas?**
> All the questions mentioned by the reviewer exhibited activation patterns that matched both body and face selective areas. It is important to underline that we do not expect a sharp distinction in the activations when referring to face and body. Neuroscientific studies have shown that face- and body-selective areas are anatomically close in the human (and monkey) high-level visual cortex and often co-activate when a whole person or animal is present (see for example Taubert et al., 2022). We cite directly from Taubert et. al: “The vast majority of research revealing the functional profiles of face and body regions, as well as their location, have used static images or dynamic stimuli in which the face and body are cropped separately from each other. This act of disassembling whole people or animals in this way is a peculiar practice that is unique to the study of body parts compared with other objects. As emphasized in the introduction, under normal circumstances faces and bodies are seen together.”
> Our findings are consistent with this: for many of the questions related to bodies, we observed overlap with both face and body regions. For example, in Figure 4 of the submitted manuscript, the questions “Is the subject sitting?” and “Is the subject running?” show strong overlap with body-selective regions but also notable overlap with face areas. In the revised manuscript, if accepted, we will include a broader set of activation maps to better illustrate this phenomenon, as well as a more detailed explanation of the overlapping body/face activations.
>
> Taubert J, Ritchie JB, Ungerleider LG, Baker CI. One object, two networks? Assessing the relationship between the face and body-selective regions in the primate visual system. Brain Struct Funct. 2022 May;227(4):1423-1438. doi: 10.1007/s00429-021-02420-7. Epub 2021 Nov 18. PMID: 34792643.
>
> **Why did you only look at one question with respect to face areas?**
> This point is closely related to the previous question. We will include a wider set of activation maps in the revised manuscript, if accepted. We included only one question because most of the questions presupposing a subject referred to it in its entirety., while relatively few questions specifically targeted the face. One of these face-focused questions was the one we included in Figure 4 (‘Is the subject smiling?’) but other examples are ‘Is the subject wearing a hat?’ and ‘Is the main subject looking directly at the camera?’.
>
> Finally, we thank the reviewer for mentioning Tom Mitchell’s work. We are familiar with some of his studies (Mitchell T. et al., 2004), which we also find extremely interesting and related in spirit. However, the experimental paradigm in that study differs substantially from ours. In Mitchell et al. (2004), participants in a controlled setting were explicitly engaged in tasks directly tied to the output categories, for example deciding whether a visually presented word belonged to one of a small set of predefined semantic categories. In contrast, our study focuses on naturalistic image viewing: participants passively viewed complex real-world scenes while performing a continuous recognition task, reporting if they had already seen an image during the experiment. This task is unrelated to any particular semantic feature of the scene. Despite this difference, we similarly train models to answer a broad range of  visual questions directly from fMRI. Another distinction is that Mitchell et al. (2004) used feature selection to isolate predictive voxels, whereas we train a linear model directly and interpret voxel weights for each question without explicit voxel selection. To the best of our knowledge, we are not aware of work from Mitchell’s group using semantically unconstrained visual questions. We would be happy to learn of any specific studies or question sets from Mitchell’s group that we may have overlooked, as they could provide useful context or additional comparisons for our study.
>
> Mitchell, T.M., Hutchinson, R., Niculescu, R.S. et al. Learning to Decode Cognitive States from Brain Images. Machine Learning 57, 145–175 (2004). https://doi.org/10.1023/B:MACH.0000035475.85309.1b

---

### Comment · Area_Chair_1czU · 2025-08-02
**Kindly Engage with Author Responses**

Dear Reviewers,

The authors have submitted their responses to your reviews. At your earliest convenience, please take a moment to engage with their replies. Your continued discussion and clarifications will be invaluable in ensuring a fair and constructive review process for all parties.

Thank you again for your thoughtful contributions and dedication!

Warm regards,

Your AC

---

### Decision · Program_Chairs · 2025-09-17

**Decision:**

Accept (poster)

**Comment:**

In this work, the authors propose a method for interpretable visual decoding from fMRI data. Initially, some reviewers had some concerns over the paper and gave negative recommendations. Yet, the authors did a good job during the rebuttal and and the end all the reviewers have positive opinions on the paper. The AC read through the manuscript, all reviews, the discussion, and the rebuttal. The authors are highly encouraged to improve the paper quality according to the reviewers' feedback in the camera-ready version. The AC decided to accept this submission.